# Position: Code Benchmarks Should Prioritize Rigor, Reliability, and Reproducibility

Jialun Cao [1 2]   Yuk-Kit Chan [* 3]   Zixuan Ling [* 3]   Wenxuan Wang [4 †]   Shuqing Li [3]   Mingwei Liu [5]   Ruixi Qiao [6]
Yuting Han [7]   Chaozheng Wang [3]   Boxi Yu [8]   Pinjia He [9]   Shuai Wang [1]   Zibin Zheng [5]   Michael R. Lyu [3]
Shing-Chi Cheung [1 2]

## Abstract

Code-related benchmarks play a critical role in evaluating large language models (LLMs), yet their quality fundamentally shapes how the community interprets model capabilities. In the past few years, awareness of benchmark quality has grown. Yet, after a decade-scale (2014 - 2025) survey over 672 code benchmarks, we observed *a lag between growing awareness and actual practice*. For example, in 2025 alone, the number of benchmarks that ignore code coverage when providing test cases nearly matches the total count accumulated across the previous ten years. In response, we take a clear position: **Code benchmarks must prioritize rigor in benchmark construction, reliability in evaluation, and reproducibility in release**. To operationalize this position, we introduce a code benchmark guideline HOW2BENCH with 55 checklists. Finally, our further human study also exposed that the current issues not only stem from the significant effort required, but also from a lack of awareness regarding their importance.

## 1. Introduction

> ❝ *Awareness is the beginning of action; action is the fulfillment of awareness.* ❞ — Yangming Wang (1472 - 1529)

---

∗ Equal contribution. † Corresponding author. [1]The Hong Kong University of Science and Technology [2]Guangzhou HKUST Fok Ying Tung Research Institute [3]The Chinese University of Hong Kong [4]Renmin University of China [5]Sun Yat-Sen University [6]Chinese Academy of Sciences, Institute of Automation [7]Beijing Language and Culture University [8]Lero the Research Ireland Centre for Software, University of Limerick [9]The Chinese University of Hong Kong, Shenzhen. Correspondence to: Jialun Cao <jialun-cao@ust.hk>, Wenxuan Wang <jwxwang@gmail.com>.

*Proceedings of the 43rd International Conference on Machine Learning*, Seoul, South Korea. PMLR 306, 2026. Copyright 2026 by the author(s).

Large Language Models (LLMs) are increasingly evaluated on code benchmarks such as SWE-Bench (Jimenez et al., 2024) to measure their code generation, code reasoning, and debugging capabilities. These benchmarks play a critical role in shaping the understanding of LLMs' true capabilities and limitations. However, **the validity of conclusions drawn from these code benchmarks depends on the rigor, reliability, and reproducibility** of the benchmarks themselves. When the benchmark construction is flawed, the validity of conclusions may not hold.

Encouragingly, awareness of benchmark quality has grown. Guidelines (Wu et al., 2025) and enhancing approaches (Liu et al., 2023b; Qiu et al., 2024b; Yadav et al., 2024b) were proposed to provide the good practices and enhance the quality in certain aspects (*e.g.*, code coverage (Liu et al., 2023b), data contamination (Cao et al., 2024b)). As a result, *one might expect that benchmark construction practices have already become more rigorous over time.*

However, our findings suggest the opposite. Through a large-scale survey of 672 code benchmarks across the last decade (2014 – 2025), we uncover a concerning trend: **despite rising awareness of benchmark quality, the number of flawed benchmarks has continued to grow**. For example, in 2025 alone, the number of benchmarks that *ignore code coverage* when providing test cases nearly matches the total count accumulated across the previous ten years. More **empirical evidence** includes:

● 46% benchmarks did not go through *quality assurance check*; 79.8% did not consider data contamination; 67.1% did not deduplicate the data points;

● 85.0% did not ensure a *reliable judgement*, such as ensuring a high code coverage when test suites are provided; 66.1% benchmark evaluation was one-pass, without repeating the experiment to avoid randomness;

● 38.2% of the benchmarks did not provide the essential information (*e.g.*, prompts) for *reproducibility*; 80.0% did not provide the log information on the benchmarks; 14.7% are *not open source*, 2.2% only partially released;

In response, we take a clear position: **Code benchmarks must prioritize rigor in benchmark construction, reliability in evaluation, and reproducibility in release as first-class objectives.**

To operationalize this position, we introduce **a comprehensive code benchmark guideline, HOW2BENCH,** comprising 55 rigor-oriented checklists, each annotated with its relative threat level to the benchmark validity. This checklist covers the entire lifecycle of benchmark development, from design and construction to *evaluation*, analysis, and release, as shown in Figure 1.

Upon the 55-item checklist, we revisited 672 code benchmarks, and quantified longitudinal trends in how benchmark practices have evolved. We observe **a lag between growing awareness and actual practice**. Although the yearly proportion of benchmarks that prioritize data quality and evaluation credibility has increased, the absolute number of flawed benchmarks continues to rise, largely due to the rapid overall growth in benchmark production.

In addition to these concerning trends, we also identify several **positive signals**:

● During *Benchmark Design*: recent benchmarks **increasingly focus on practical, real-world problems**; notably, the number of project-level benchmarks in 2025 nearly tripled compared to 2024, and the distribution of code task types has become more diverse;

● During *Benchmark Construction*: creators are showing **stronger awareness of manual quality assurance**: the number of benchmarks incorporating human verification more than doubled from 2024 to 2025;

● During *Benchmark Evaluation*: benchmark creators increasingly include a larger and more diverse set of studied LLMs, improving the generalizability and robustness of reported findings;

● During *Benchmark Release*: growing open-science engagement, with **an upward trend in publicly released artifacts, prompts, and evaluation resources**, reflecting stronger community commitment to transparency and reuse.

**Human Study** – To better understand whether the current situation stem from lack of awareness, resource constraints, or incentive misalignment, we conducted a human study involving 49 participants through questionnaires. All participants concurred on the necessity of having a checklist for benchmark construction to enhance quality. Interestingly, beyond admitting the its importance, the collected feedbacks also exposed *gaps in awareness*: 16% of participants were unaware of the necessity for data denoising; over 40% were not aware that the experimental setup and environment could impact the reproducibility and transparency.

The questionnaires results revealed that the current issues in flawed benchmark not only stems from the significant effort required to construct a rigorous and reliable benchmark, but also from a lack of awareness regarding how crucial a rigorous development process is to the quality of benchmarks and the reliability of evaluations. Without being aware of and addressing the quality of code-related benchmarks (as well as other kinds of LLM benchmarks), the reliability and reproducibility of benchmark results remain compromised, misleading research directions, and hindering meaningful progress.

This paper makes contributions in five aspects:

- *Novelty*. We introduce HOW2BENCH, a comprehensive set of guidelines packaged as a 55-criteria checklist that covers the lifecycle of code-related benchmark development.

- *Significance*. HOW2BENCH presents the first comprehensive set of actionable guidelines for developing high-quality benchmarks, striving to create a more reliable and transparent environment. The human study also highlighted the demand for such a detailed guideline.

- *Usefulness*. HOW2BENCH serves as a guideline for practitioners before/during developing code-related benchmarks, and a checklist for evaluating existing benchmarks after their release. For ease of use, we also provide a *printable version* of HOW2BENCH on Appendix G.

- *Generalizability*. Most criteria listed in HOW2BENCH can be adopted or adapted to other benchmarks such as Question-answering, mathematical reasoning, and multi-modal benchmarks.

- *Long-term Impact*. Our statistics alert the community to the severity and prevalence of non-standard practices in benchmark development. It ultimately improves the overall quality of benchmarks through propagation effects among them.

## 2. Related Work

### 2.1. Code Benchmarks

Benchmarks for coding tasks like code generation (Chen et al., 2021a; Austin et al., 2021), defect detection (Just et al., 2014; Gao et al., 2023b; Liu et al., 2024d), and program repair (Jimenez et al., 2024; Risse & Böhme, 2024) are increasingly common, reflecting the growing needs for using LLMs for coding tasks. Recent studies have highlighted various issues with these benchmarks, ranging from design inconsistencies to scope and applicability limitations. For example, (Liu et al., 2023b) found that even some widely used benchmarks, such as HumanEval (Chen et al., 2021a)

and MBPP (Austin et al., 2021), contains a non-trivial proportion of bugs in implementation, documentation, and test cases. Our work, in comparison, introduces a detailed guideline that **_guides the benchmark development_** during the entire lifecycle.

## 2.2. Benchmark Surveys and Studies

Several recent surveys (Koohestani et al., 2025a) and empirical studies have profiled the status quo of LLM development. These studies either explore the overall performance for certain areas, such as software engineering (Hou et al., 2023; Wang et al., 2024a), or investigate the capabilities of LLMs on specific tasks, such as code generation (Dou et al., 2024; Yu et al., 2024) and test generation (Schäfer et al., 2024; Yuan et al., 2024b; 2023b). A survey (Chang et al., 2024) about how to evaluate LLMs was proposed to answer what/where/how to evaluate LLMs. This paper differs from these studies in its purpose and perspectives.

Assessments or guidelines for benchmarks (Qian et al., 2026; Koohestani et al., 2025a; Hu et al., 2025b; Reuel et al., 2024) are also constantly released. For example, BetterBench (Reuel et al., 2024) is a related work assessing the AI benchmarks against 46 criteria. Then, it scored 24 AI benchmarks in various domains and ranked them. BetterBench differs from this paper in several key aspects: scope (general benchmarks vs. code-related benchmarks), lifecycle division (it addresses benchmark retirement, while How2Bench focuses on benchmark evaluation, analysis, and release), and objectives (scoring benchmarks vs. offering comprehensive guidelines for future benchmark development). Additionally, the study in this paper was conducted on a much larger scale (24 vs. 572 benchmarks), statistically highlighting the prevalent issues in existing benchmarks.

Unlike these surveys and guidelines, our work reveals a longitudinal trends before and after them came out.

## 3. Guideline Design

### 3.1. The Lifecycle of Benchmark Development

Code-related benchmark development comprises five typical phases (Phase 1 - 5), as shown in Figure 1, explained in detail as follows.

**Phase 1. Design**. At the beginning of benchmark development, it is vital to identify the motivation, the **_scope_** and the **_capabilities_** required by the **_application_** scenario of interest. To achieve this objective, one needs to carefully consider the application scenarios, making sure these scenarios align with real-world demands (Moriarty, 2011). Also, it is necessary to assess whether other benchmarks already exist that address similar tasks, and to identify any shortcomings they may possess (McIntosh et al., 2025; Malode, 2024).

Furthermore, this new benchmark should be designed to evaluate specific LLMs' capabilities; the crafted tasks are expected to reflect these capabilities (Hodak et al., 2023).

**Phase 2. Construction**. Benchmark Construction phase moves from design to execution. Typically, data is **_collected_** from public coding websites such as GitHub, LeetCode, and StackOverflow. This is followed by preprocessing, which includes filtering, cleaning (e.g., deduplication, denoising), and **_curation_** (e.g., aligning tests with corresponding code). The phase usually ends with a **_validation_** process, which can be manual or automated (McIntosh et al., 2025).

**Phase 3. Evaluation**. Once the benchmark is available, the next step is to apply it to LLMs, validating if it can effectively measure the intended LLM capabilities. Essential considering factors include **_selecting a representative array of LLMs_**, configuring **_settings_** like prompts and hyperparameters for consistency, choosing appropriate experimental **_environments_** to meet LLM requirements, and implementing thorough **_logging_** to ensure dependable and reproducible results.

**Phase 4. Analysis**. After evaluation, experimental results are analyzed, drawing conclusions on LLMs' capabilities. This phase involves comparing each LLM's **_performance_** to identify standout or underperforming models. Then, proper visual aids such as bar charts and tables can be used to **_display_** the experimental results, presenting clearer observation and deeper **_inspiration_**, such as the correlations between models, the correlations with related benchmarks, or performance in upper-/down-stream tasks (Hendee & Wells, 1997). Indeed, a thorough analysis helps pinpoint areas for improvement and guides future LLM enhancements.

**Phase 5. Release**. The final phase is to make the benchmark open-accessible. This phase involves meticulously preparing all **_materials_** associated with the benchmark, ensuring they are ready for **_open access_** to foster widespread adoption and collaboration. Clear, comprehensive **_documentation_** is provided to guide users on effectively utilizing the benchmark. Additionally, all logged **_experiment details_** are made available, enhancing the reproducibility and transparency of the benchmark.

### 3.2. Study Design

Our study consists of four steps (Figure 2). All steps are explained as follows.

**Step 1. Guideline Construction**. To begin with, we **_sketched the initial guidelines_** for each phase in the benchmark development lifecycle (Section 3.1, Figure 1) by reviewing existing literature (Suppes et al., 1962; Zheng et al., 2023b; Schäfer et al., 2024; Reuel et al., 2024) and brainstorming. After that, we **_refined the guidelines_** through a series of interviews with various stakeholders, includ-

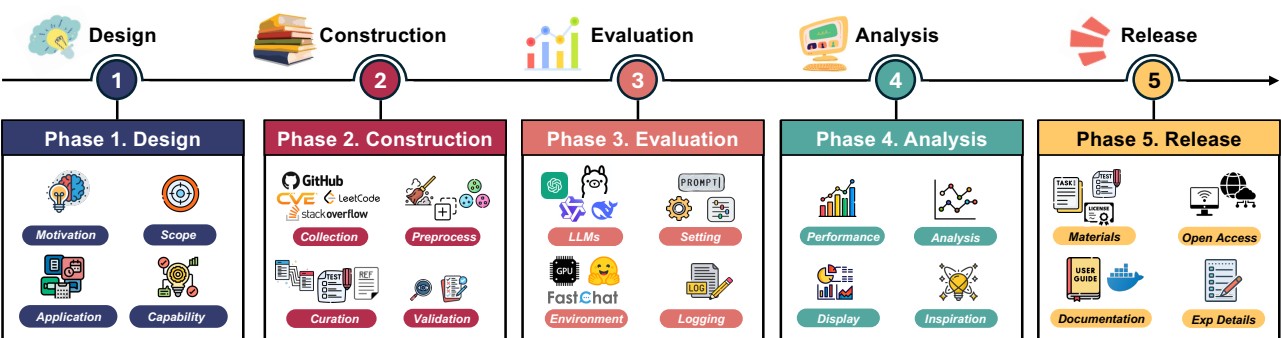

*Figure 1.* Lifecycle of Benchmark Development

ing model developers and benchmark builders, allowing for the addition, deletion, or modification of criteria based on expert feedback and practical insights. In order to allow more flexibility and increase the practicality of the guideline, we prioritized them with ★★★ (Highly important), ★★ (Important), and ★ (Optional). By doing so, the highly-recommended criteria are essential to comply with when setting up a new benchmark, while other criteria allow compromise, making the guideline simpler for benchmark developers to follow.

**Step 2. Literature Profiling**. This step begins by *collecting related benchmarks* according to their publication time, venue, and coding tasks, then employing techniques like *snowballing* to ensure a comprehensive collection. This step leads to 672 code-related benchmarks for study. The detailed statistics are available in Appendix F. This step is followed by *profiling* each selected benchmark through a thorough review of *corresponding papers* and examination of *the released artifacts* or homepages associated with these benchmarks. The phase is completed by *reporting statistics* that highlight overall trends, pros, and cons identified during the profiling, providing a structured overview of existing benchmarks.

**Step 3. Focused Case Study**. After obtaining an overall impression of existing benchmarks, we *selected 30 (= 5 * 6) representative benchmarks* from the top-5 most frequent coding tasks (see Figure 9), with top-5 highly-cited benchmarks plus the latest 1 benchmark (Appendix E). Each selected benchmark is then *analyzed against* HOW2BENCH, examining how well they meet the established criteria, studying their overall statistics, and identifying both exemplary and poor cases. Insights and references from existing literature are also incorporated to enrich the analysis, providing a deeper understanding of the benchmarks' performance and areas for improvement.

**Step 4. Human Study**. The final step is a human study that evaluates the importance and practicality of HOW2BENCH. This involves *designing a questionnaire* by first initiating

and iterating to gather diverse, logical insights, which is then *distributed* to a targeted audience. After collecting and filtering responses for quality, the data is *analyzed* to derive insights. See Appendix D for details.

## 4. Guideline and Key Statistics

The completed guideline HOW2BENCH with 55 criteria can be found in the Appendix. Each guideline contains: an actionable check criterion with necessary explanations; a priority indicator, divided into three levels in total, marking the importance of the checklist; and a checkbox for convenience.

### 4.1. Guideline for Benchmark Design

*Explanation –* For benchmark design, we listed four essential criteria, as shown in Figure 3. In particular, the guideline starts by recommending that benchmarks should initially assess if they are addressing a *significant gap* in existing research, ensuring the relevance and necessity of the benchmark. The *scope* of the benchmark is expected to be well-defined, clarifying the *capabilities* or characteristics being tested, how these relate to practical scenarios such as programming assistance or automated testing, and the relevance of these capabilities in real-world *applications*.

■ *Key Statistics –* According to our statistics among 672 benchmarks, *apparent research bias* can be observed in terms of coding tasks, programming languages, and code granularities (Appendix C.1). For example, *Code Generation* is most prevalent coding task, with 235 benchmarks focusing on this area, according to 34.97% (235/672) of studied benchmarks, indicating a significant interest in generating code automatically. The second most prevalent is code reasoning 12.64% (85/672), followed by Program Repair and Defect Detection.

When examining the distribution of coding tasks by year (see Figure 10), we observe that benchmark growth accelerated most rapidly in the past two years (2024–2025),

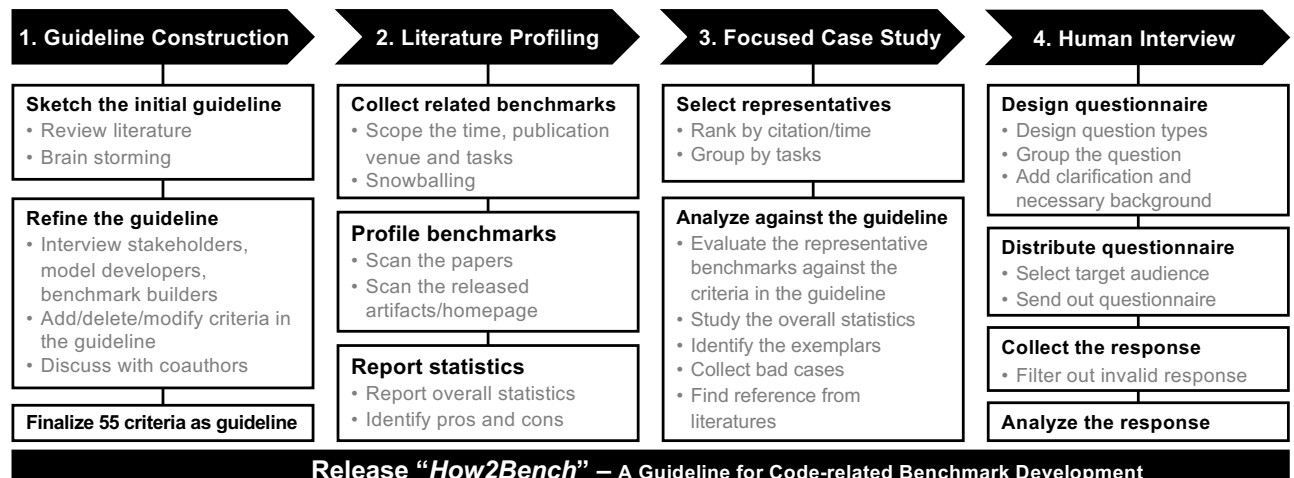

*Figure 2.* Workflow of study process

with code generation remaining the most frequently benchmarked task. At the same time, **demand for code reasoning has increased substantially**, rising from 19 benchmarks in 2024 to 61 in 2025. Benchmarks for program repair also increased over the same period, from 22 to 33.

Regarding the bias in programming language, Figure 11 shows that 409 (71.50%) benchmarks are in Python, followed by Java and C++, with 229 and 160, respectively. This observation consolidates the observation from previous works (Cao et al., 2024a; Hou et al., 2023) on a larger scale. When analyzed by year (Figure 12), the number of benchmarks for C/C++ and JavaScript has increased noticeably between 2024 and 2025. We also observe a sharp rise in Rust-related benchmarks: although only a few benchmarks existed for Rust since its release in 2015, 57 benchmarks were published in the past two years (27 in 2024 and 29 in 2025). These trends in programming language coverage reflect the evolving demand for LLMs across different languages, highlighting particularly the growing reliance on large models to generate, reason about, and maintain code in these languages.

Regarding the bias in granularity, a dramatic upward trend was observed in project-level benchmarks (Figure 16): between 2024 and 2025, the number of project-level benchmarks surged, increasing from 35 to 97 new benchmarks. Importantly, these numbers refer to benchmarks newly introduced in each year, rather than cumulative totals. This surge indicates a **growing community focus on the real-world applicability and large-scale practical utility of LLMs**.

Regarding the LLMs' capability evaluated, during the focused case study (listed in Appendix E), we identified that 20% benchmarks have not explicitly specified the capabilities (e.g., intention understanding, program synthesis) to

be evaluated, and ***23.3% have not specified application scenarios*** the benchmark targets.

Besides, we also identified a case in MBPP (Austin et al., 2021) where a case fell out of the target evaluation capabilities (Appendix C.2). Indeed, clearly defining the application scenarios/scopes/capabilities could help benchmark constructors establish precise goals for the design and development of the benchmark, ensuring accuracy in the evaluation.

> ▲ **Severity** – Current benchmarks exhibit ***an apparent imbalance*** in coding tasks and programming languages dominated by code generation and Python, leaving research blank to be filled. Also, even highly cited benchmarks may have samples that do not fall into the examined capabilities.

### 4.2. Guideline for Construction

☞ ***Explanation*** – Figure 4 shows 19 criteria for benchmark construction. Essentially, for **data source**, the key considerations include verifying the traceability and quality of the data source, addressing potential *data contamination* (Sainz et al., 2023), and ensuring that the ***data sampling processes*** are scientifically robust and rigorous. Also, for **data representativeness**, it also guides through specific checks to ensure the benchmark's scope is strictly adhered to, such as making sure every data point falls within the targeted scope and that the data can cover all studied capabilities, domain knowledge, and application scenarios.

For data preprocessing and cleaning, it also stresses handling code-specific aspects, such as compilability and execution, along with cleaning and ***manually reviewing*** data for quality assurance. Output validation methods and evaluation metrics must be carefully designed and reviewed to ensure

they effectively measure the benchmark's goals. Lastly, it suggests considering additional evaluation perspectives, such as safety (Wei et al., 2024; Yuan et al., 2024a) checks, ensuring the code does not contain sensitive information.

■ *Key Statistics* – According to our statistics (Appendix C.3), the 572 benchmarks exhibit **numerous irregularities** in their implementation, which could significantly threaten the reliability of the benchmarks. Surprisingly, 67.1% of benchmarks did not deduplicate or did not mention. 79.8% benchmarks did not consider or handle data contamination threats. About 46.0% of the benchmarks did not go through any quality assurance checks such as manual checks and code execution. In particular, we summarized the *commonly-used data quality assurance metrics* and their frequency: manual check (45.8%), code execution (1.2%),others (*e.g.*, heuristic rules, 6.4%).

Examining the trends by year (Figure 24), however, **one positive development is the increasing prevalence of manual quality checks**. The number of benchmarks with manual check guarantees doubled from 84 in 2024 to 193 in 2025, indicating that a growing share of code benchmarks now undergo partial or full human verification.

Also, since we focus on code-related benchmarks, which usually accompany test cases, *code coverage* also needs to be considered. According to the statistics over oracles (Figure 31), **passing test cases** (257 / 672 = 38.2%) and **the exact match** (193 / 672 = 28.7%) are the most common oracles used in code benchmarks. However, we observed that only 15.9% considered and reported code coverage (*i.e.*, line coverage, branch coverage) explicitly in their papers, while 82.5% did not consider the code coverage when constructing the test suites for benchmark evaluation. The annual distribution (Figure 33) makes this trend even clearer: although many benchmarks in the past three years (2023–2025) did not consider code coverage, the sheer volume of benchmarks in 2025 means that the absolute number of such benchmarks is high, with 24, 77, and 123 benchmarks respectively for 2023, 2024, and 2025. This underscores that, despite growing awareness of evaluation rigor, a substantial number of benchmarks continue to provide incomplete testing, highlighting a persistent threat to the validity and reliability of benchmark-driven assessment. It severely affects the reliability of findings on these benchmarks, potentially misguiding future research and applications based on these flawed assessments.

> ▲ **Severity** – Most benchmarks display *severe loopholes* in data preparation and curation, *i.e.*, only 54% of benchmarks went through a quality assurance check, and only 20.2% of them considered and handled data contamination threats.

## 4.3. Guideline for Evaluation

☞ *Explanation* – Guidelines for benchmark evaluation focus on the rigorousness and reliability of the evaluation. HOW2BENCH provides 12 criteria for benchmark evaluation, as shown in Figure 5. It mainly focuses on the comprehensive evaluation processes for benchmarks involving LLMs. For evaluation design, it stresses the importance of assessing a sufficient and *representative range of LLMs* to ensure the benchmark's applicability across various model families and configurations, both open and closed-source. Figure 35 and Figure 37 show the distribution of numbers of LLMs studied and the most exercised LLMs.

Also, *prompting* has a direct impact on the quality of the LLMs' output results (Wei et al., 2022; He et al., 2024a; Jin et al., 2024; Ye et al., 2023). As pointed out by a recent study, up to 40% performance gap could be observed in code translation when prompts vary (He et al., 2024b).

Additionally, *the experiment environment* is essential for reproducibility and transparency. Indeed, the hardware, software, and platform environments used during experiments might influence the outcomes (Zhang & Huang, 2025). Furthermore, because of the nondeterministic nature of LLMs, experiments should be repeated, and randomization strategies should be used to mitigate the effects of randomness and parameter configuration biases. Lastly, *meticulously documented logs* of the experimental process is advised to facilitate transparency and reproducibility, detailing everything from parameter settings to the specific LLM pipelines such as vLLM (Kwon et al., 2023) used.

■ *Key Statistics* – Among the 672 benchmarks, 585 of them are evaluated over LLMs. As shown in Figure 35, most benchmarks were evaluated against six LLMs (10.0% = 59 / 585), followed by six LLMs. Encouragingly, increasing LLMs are being studied for code benchmarks. As shown in Figure 36, in 2024, 39.5% of benchmarks were evaluated against fewer than five LLMs, whereas in 2025, 64.2% of benchmarks (36.7% + 27.5%) were evaluated against 5 - 20 models. This shift indicates that benchmark studies are increasingly aiming for more comprehensive and generalizable evaluations. However, it also introduces substantially higher computational and financial costs, highlighting the trade-off between evaluation rigor and resource efficiency.

For reference, we listed the top 10 most studied LLM families in Figure 37. Among them, the GPT series from OpenAI is the most extensively studied, accounting for 76% (446/585), followed by Deepseek and Qwen.

The prompt quality also significantly impacts the LLM evaluation (He et al., 2024b). According to a recent study, up to 40% of performance variation could be observed in the code translation task (He et al., 2024b). So, carefully designing a prompt needs consideration. However, **76.7%** represen-

tative benchmarks (Appendix E) do not validate whether the prompts they used are well-designed (Appendix C.4). Similarly, though 89.4% benchmarks were evaluated in a zero-shot manner, only 18.6% benchmarks were evaluated under few-shot, 3.7% under Chain-of-Thought and 1.0% under RAG (Figure 40). However, as shown in Figure 41, 76.7% representative benchmarks (Appendix E) do not validate whether the prompt they used is well-designed.

Regarding the evaluation process, our statistics exposed that **only 33.4% of benchmark evaluations have been repeated** (Appendix C.4). Also, regarding the transparency and matriculated documents, the observation is not optimistic – **Only 6.3% benchmarks provided their experiment environment**. **More than 38% of benchmarks did not provide reproducible instructions** such as prompts, examples for few-shot learning, or content for retrieval (Figure 48). **Only half (50.5%) provide hyperparameters** such as temperature for reproduction.

> ▲ **Severity –** 66.6% of evaluations have not been repeated to eliminate the impact of randomness, and the trend grows from the year 2023 to 2025 (from 31 to 205).

## 4.4. Guideline for Evaluation Analysis

☞ *Explanation –* The analysis of the experiment results is expected to be objective and comprehensive, hopefully providing insights or actionable advice. So, we listed 10 criteria for the evaluation analysis phase, as shown in Figure 6. Regarding **the perspectives of analysis**, inspired by classic measurement theory (Suppes et al., 1962), we suggest four essential perspectives, including *difficulty* (whether a benchmark is appropriately challenging for LLMs), *stability* (whether the results are consistent through repeated trials), *differentiability* (whether benchmarks can differentiate the strengths and weaknesses of various LLMs), and *inspiration* (e.g., the correlations between the upper-/down-stream coding tasks and LLM scores).

Moreover, effective *presentation of results* using clear visual and textual descriptions could ensure the findings are understandable and actionable. The phase concludes with the suggestion to interpret and explain the results comprehensively, providing a basis for future enhancements.

■ *Key Statistics –* Because experimental analysis is relatively subjective and cannot be obtained through mechanical scanning, we focus on 30 representative focus benchmarks (Appendix E), covering the highest cited and latest benchmarks in top-5 tasks. Figure 45 shows an example from CruxEval (Gu et al., 2024) where the experimental scores can hardly be read from the figures.

Also, *explaining experiment results* is crucial for other practitioners to understand what the outcomes mean in the context of the research questions. According to our statistics (Appendix C.5), 70% of benchmarks have detailed explanations and analyses of their evaluation results, while still ***30% have not***. Indeed, an explanation contributes to the body of knowledge by making it possible to understand and compare results with previous studies, promoting transparency within the community.

> ▲ **Severity –** The analysis of experimental data and the clarity of data presentation may receive less attention and be worth consideration. Even in papers cited 2k+ times like MBPP (Austin et al., 2021), there are instances of*unclear evaluation analysis and display*.

## 4.5. Guideline for Benchmark Release

☞ *Explanation –* Finally, releasing a benchmark for open access also needs careful consideration. We offered 10 suggestions for this step, as shown in Figure 7, to highlight essential steps for public release preparation, emphasizing accessibility and ethical compliance. This includes setting an appropriate **license** to clarify usage rights, conducting a thorough review to **eliminate sensitive or harmful content** such as the API keys to access LLMs, the personal emails or toxic code comments (Miller et al., 2022) unless they are a part of the benchmark, and ensuring reproducibility by making all related materials openly available. **Detailed prompts** and clear descriptions of the experimental setup are advised to facilitate replication. Additionally, providing user manuals and evaluation interfaces is crucial for effective user engagement with the benchmark, enhancing its reliability and value for the research community.

■ *Key Statistics –* The final step involves the release of the benchmark. The fundamental requirement for releasing a benchmark is that it must be open-sourced. However, surprisingly, we observed that 2.2% of the benchmarks are only partially open-sourced (e.g., missing some subjects or tests), and ***14.7% are not open-sourced at all*** (e.g., links/web pages are no longer active). Furthermore, prompts, which are necessary for reproducibility, are not disclosed in 38.2% of the benchmarks (Figure 48), not to mention the lack of public information on experimental settings (Figure 39 and Figure 38) and experimental parameters (Figure 53). What is worse, 19.3% benchmarks do not set up licenses (Figure 54). The absence of licensing may lead to severe legal and ethical issues, potentially resulting in unauthorized use and distribution of proprietary technologies. Additionally, only 16.7% of the benchmarks make their logged experimental results publicly available (Appendix C.6). Note that this conclusion about log availability are based on publicly accessible materials; logs are often missing for a combination of policy, privacy, and practicality reasons.

Fortunately, the recent years show a positive shift toward

openness (Figure 47). From 2024 to 2025, the number of open-sourced benchmarks increased substantially, rising from 169 to 266, indicating growing community commitment to transparency, accessibility, and reproducibility.

> ▲ **Severity –** The release of existing benchmarks exhibits several issues. For example, 16.8% of the benchmarks are either not open to public access or are only partially open-sourced. Only 47.4% of benchmarks are released with replicable prompts.

## 5. Human Study

To delve deeper into the integration of knowledge and action, we *surveyed 49 global researchers* in AI (42.6%) and Software Engineering (57.14%), as shown in Figure 61. Each participant had published at least one research paper to ensure their research maturity, and *half had constructed code benchmarks*. See Appendix D for more details about the participants' demographics and questions in questionnaires.

First, **all participants agreed** that having a checklist for benchmark construction would contribute to the quality of the benchmark. 47/55 criteria in HOW2BENCH are deemed important by more 80% participants. Additionally, among the 21 participants who have constructed code-related benchmarks, *53 out of 55 criteria were deemed important by all benchmark developers*; only two criteria (criteria 3 and 4 in Section 4) were considered unimportant by a few individuals (3 and 2 participants, respectively). Additionally, we received two valuable suggestions that draw importance to recording *the time/monetary costs* of constructing the benchmark and conducting the experiments.

However, we also identified some *notable gaps in awareness*. First, regarding the *data preparation*, more than 15% of participants were not aware that the selection of data should consider the target scope of the evaluation set (i.e., the data must be representative). 16% of participants were *unaware of the need for data denoising*, while half (8%) of these have already published at least one paper on benchmarking construction. This oversight can significantly affect the validity and generalizability of experimental results, underscoring the importance of a comprehensive understanding of data handling for reliable research outcomes.

Second, regarding *evaluation replicability and reliability*. Over 40% of participants believe that recording and publicizing the hardware and software environments, software versions, and libraries used in experiments is not important, with more than 20% still considering it unimportant despite already done so. This reveals *a significant lack of awareness* about the impact that experimental environments can have on *the reliability, reproducibility, and stability of evaluation results*. In fact, various studies have demonstrated

that different experimental environments, parameters, and prompts can lead to substantial variations in outcomes (Xiao et al., 2024; Wang et al., 2019; 2023a).

## 6. Alternative Views

We clarify several seemingly natural but ultimately misleading ways to frame our work. **Alternative View 1: Frame our work as a survey of code benchmarks**. Our decade-scale analysis of 672 benchmarks may suggest that this paper is primarily a survey. While we do provide quantitative and qualitative observations, our main contribution is normative and prescriptive, not purely descriptive. Rather than cataloging existing work, we (i) expose systematic gaps between awareness and practice, and (ii) propose actionable standards and tooling directions for future benchmark design. **Alternative View 2: Frame our work as a criticism of specific benchmarks or communities**. Our intent is not to single out particular benchmarks or communities as bad. The issues we study are widespread and systemic. We emphasize structural incentives and use HOW2BENCH to offer a constructive, forward-looking path for community-wide improvement. **Alternative View 3: Frame our work as a call for perfectionism or over-engineered benchmarks**. Prioritizing rigor, reliability, and reproducibility does not mean that all checklists must be satisfied for a benchmark to be "acceptable". HOW2BENCH is a structured checklist and prioritization tool, not a rigid standard. It makes trade-offs explicit, helps authors identify the most critical gaps, and supports transparent choices. Our goal is to shift the default toward more principled practices and clearer documentation, not to demand maximal rigor in every dimension.

## 7. Discussion

### 7.1. Trade-offs Between Benchmark Rigor and Development Efficiency

Despite all the above arguments, we admit that constructing rigorous benchmarks often entails a significant investment of time and human effort, which can lead to reduced efficiency in the development and evaluation process. Indeed, ensuring data quality, implementing thorough validation procedures, and designing comprehensive evaluation protocols requires non-trivial effort and may slow down the pace of research. It also echoed our human study. Through our human study, we found that researchers are often aware of the importance of several criteria (*e.g.*, data denoise and repeating the experiments), but did not implement them due to time constraints or other limitations.

However, this trade-off between rigor and efficiency is necessary to guarantee the reliability, reproducibility, and scientific value of benchmark results. While faster, less rigorous

benchmarks might accelerate short-term experimentation, they risk producing misleading or non-generalizable findings that ultimately hinder long-term progress. Therefore, each checklist in HOW2BENCH is labeled with a **priority**: ★★★ (Highly important), ★★ (Important), and ★ (Optional). We hope the design of this indicator may better help benchmark developers balance efficiency with rigor.

## 7.2. Awareness and Action

It is also worth recapping the notable gaps in awareness of the importance of data preparation and reproducibility (Section 5). For example, 16% of participants were unaware of the need for data denoising; 40% of participants believe that recording and publicizing the hardware and software environments, software versions, and libraries used in experiments is not important. However, without being aware of and addressing the quality of LLM benchmarks, the reliability and reproducibility of benchmark results remain compromised. This observation should be a call to action for the research community to strengthen education and awareness around best practices in benchmark development.

## 7.3. Criteria on Literature Labeling

**Labeling Process**: Because the literature scanning and annotation (*i.e.*, Step 2 in the study workflow) involve manual labeling, we summarize the concrete steps we followed when labeling the 672 benchmarks. First, **Independent annotation by two primary annotators**. For each benchmark paper, two primary annotators independently: (1) Skimmed the full paper and then carefully read sections likely to contain relevant information (e.g., Dataset/Data, Data Collection, Data Curation/Cleaning, Quality Control/Assurance). (2) Assigned each attribute a label from present, not mentioned. (3) Recorded textual evidence (exact sentences or locations) whenever an attribute was labeled as "present", storing all evidence in a shared annotation sheet. Second, **Conflict detection and resolution with additional annotators**. After both primary annotators completed their labels, we automatically identified conflicts (e.g., one annotator labeled "has contamination check" as present while the other labeled it as absent). For each conflicted benchmark–attribute pair: The two primary annotators first re-checked the paper and their evidence. If disagreement remained, two additional annotators joined the discussion to reach a consensus. Finally, **Finalization and quality control**. Only consensus labels were used in our descriptive statistics. We calculated inter-annotator agreement using Cohen's kappa; it ranged from 0.76 to 0.92 across attributes.

**Labeling Criteria**: The manual annotation (*e.g.*, determining whether a paper considers data denoising) follows the annotation criteria: (1) **Criteria for checking whether data quality assurance is present**: We labeled a benchmark as "considers data quality assurance" only if we identified a dedicated section, subsection, or explicit sentences describing how data quality was ensured. In the absence of such textual evidence, we labeled the benchmark as not considering data quality assurance. Annotators look for explicit discussion of how data quality is ensured. Typical evidence includes sections or sentences tagged as "Data Curation / Cleaning / Preprocessing", "Quality Control / Quality Assurance", etc., or detailed descriptions of construction and filtering procedures. (2) **Criteria for data quality assurance by manual check**: We mark a benchmark as using manual checks if it explicitly mentions human involvement in inspecting or validating data/labels, with keywords such as "manual", "human", "inspection", "verification", "review", "spot check", "random sampling", "audt", "sanity check", "hand-curated", "human-in-the-loop", "expert", "label correction", "error analysis", "case-by-case". (3) **Criteria for data quality assurance by code execution**: We mark a benchmark as using code execution checks if it describes validating items by actually running code or tests, with keywords such as "execute/execution", "run", "compile", "build", "unit tests", "test suite", "pass/fail". (4) **Criteria for data quality assurance by LLM-as-judge**: We mark a benchmark as using LLM-as-judge if it delegates evaluation or filtering to a model as a judge, with phrases like "LLM-as-a-judge", "LLM judge", "model judge", "[model name] as judge". (5) **Criteria for data contamination check**: We mark a benchmark as checking for contamination if it explicitly examines or mitigates training–test overlap. We rely on terms such as: "data contamination", "leakage", "overlap", "Jaccard similarity", "time-based split", "cutoff date", "post-YYYY". (6) **Criteria for checking Data deduplication**: We mark deduplication as present if the benchmark explicitly states with cues like: "deduplication", "de-dup", "remove duplicates", "duplicate removal", "redundant", "repetitive". (7) **Criteria for checking coverage considerations**: For code coverage, we look for mentions of: "coverage", "test adequacy", "test strength", "test suite quality", 'corner case(s)" or related phrasing.

## 8. Conclusion

Awareness of benchmark quality has grown. Yet, we observed a lag between growing awareness and actual practice after surveying 672 code benchmarks. This position paper argues that code benchmarks should prioritize rigor in benchmark construction, reliability in evaluation, and reproducibility in release as first-class objectives; users should actively scrutinize the quality and credibility of the evidence the benchmark designer provides for each claimed category. To operationalize this position, we introduce a code benchmark guideline HOW2BENCH with 55 checklists. We hope this work serves as both a warning and a catalyst, encouraging the community to adopt stronger shared standards.

## Acknowledgements

This work was supported by the National Natural Science Foundation of China (Grant No. 92582201, No. 62402113), the Hong Kong SAR Research Grant Council/General Research Fund (Ref No. 16210725), the Hong Kong SAR Research Grant Council/Theme-based Research Scheme (Ref No. T41-517/25-N), Guangdong Basic and Applied Basic Research Foundation (Grant No. 2024A1515010145), Taighde Éireann – Research Ireland under Grant Number. 13/RC/2094_2, and Research Grants Council of the Hong Kong Special Administrative Region, China (RGC Ref. No. SRFS2425-4S03 of the Senior Research Fellow Scheme).

## Impact Statement

This position paper argues that code benchmarks should prioritize **rigor** in construction, **reliability** in evaluation, and **reproducibility** in release.

**Reframing Benchmark Quality** By analyzing a decade of code benchmark development (2014–2025), we present evidence that *awareness of benchmark quality has not yet fully translated into a more rigorous practice*. We argue that benchmark quality should not be treated as an informal best effort, but as a first-class obligation. Our proposed lifecycle-aware framework and 55-item checklist articulate a concrete, actionable standard for what constitutes a methodologically rigorous, reliable, and reproducible benchmark, providing a shared reference point for researchers, reviewers, and benchmark creators.

**Shaping Community Norm** This work advocates for a shift in community norms *away from speed-driven benchmark releases and leaderboard-centric evaluation toward long-term reliability*. By making evaluation rigor more explicit and measurable with the checklist (HOW2BENCH), we aim to influence how benchmark papers are reviewed and how research contributions are rewarded.

**Broader Influence on Machine Learning Evaluation.** Although our empirical analysis focuses on code-related benchmarks, the core position advanced in this paper, *i.e.*, benchmark progress should be constrained by methodological rigor, reliability, and reproducibility, *extends to evaluation practices across machine learning*, including vision, natural language processing, robotics, and multimodal AI. We hope this work advances a broader shift and serves as a warning toward rigorous evaluation, strengthening the transparency, comparability, and trustworthiness of benchmarks.

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

# A. Appendix

The appendix is organized as follows:

- **Appendix B** lists the checklist items in each benchmark development phase.

- **Appendix C** lists all the statistics in the survey.

- **Appendix D** explains the details of human study.

- **Appendix E** lists all the benchmarks in focused study.

- **Appendix F** lists all the surveyed benchmarks in a chronological order.

- **Appendix G** shows the complete HOW2BENCH with 55 checklists.

# B. Detailed Guidance in HOW2BENCH

We present the detailed guidance in each phase.

| | Phase 1. Benchmark Design | Priority | ☑ |
|---|---|---|---|
| 1 | Consider whether the benchmark can fill the gap in related research. | ★★★ | ☐ |
| 2 | Consider what is the expected scope of the benchmark set (e.g., what natural languages, programming languages, task granularity). | ★★★ | ☐ |
| 3 | Consider the expected application scenario of this benchmark (e.g., programming assistant, automated tester). | ★★★ | ☐ |
| 4 | Consider the LLMs' capabilities (e.g., understanding, reasoning, calculation) and domain knowledge (e.g., OOP, memory management, fault localization, process scheduling) that the benchmark hopes to evaluate. | ★★★ | ☐ |

*Figure 3.* **Guideline for Benchmark Design**

| | Phase 2. Benchmark Construction (1/2) | Priority | ☑ |
|---|---|---|---|
| 5 | Consider whether the data source of the benchmark is traceable. | ★ | ☐ |
| 6 | Consider whether the data source of the benchmark is of high quality (e.g., stars, downloads, last update times, number of forks). | ★★ | ☐ |
| 7 | Consider whether the benchmark's data source is representative (e.g., choose an open-source community or code hosting platform that matches the task, capability, and scope under study) | ★ | ☐ |
| 8 | Consider data contamination issues during the benchmark collection (e.g., considering the upload time of the source code or checking whether the data source is included in the training data of LLMs). | ★ | ☐ |
| 9 | If data sampling is needed, consider whether the choice of sample size is scientific (e.g., considering the confidence level/margin of error/sampling proportion, etc.). | ★ | ☐ |
| 10 | If data sampling is needed, consider whether the sampling process is rigorous (e.g., random sampling, stratified sampling, etc.). | ★ | ☐ |
| 11 | Ensure each data point in the benchmark falls into the targeted scope (e.g., checking each data point's evaluated capabilities or domain knowledge). | ★ | ☐ |
| 12 | Consider whether the data in the benchmark can cover the studied capabilities/domain knowledge/application scenarios. | ★★★ | ☐ |
| 13 | Consider whether there is a standard answer for each sample in the benchmark (such as reference code, etc.). | ★★ | ☐ |
| 14 | For code, consider whether the code is compilable/executable. | ★ | ☐ |
| 15 | Consider the possibility of noise in the data and perform denoise. | ★★ | ☐ |
| 16 | Consider the possibility of duplication in the data and deduplicate them. | ★★ | ☐ |
| 17 | Clean the sensitive information (such as data desensitization and anonymization) unless the benchmark is deliberately designed so. | ★★ | ☐ |
| 18 | Manually review some or all of the data in the benchmark to ensure its quality. | ★★ | ☐ |
| 19 | Use LLMs to review some or all of the data in the benchmark to ensure its quality. | ★ | ☐ |
| 20 | Design appropriate output validation methods for the benchmark (e.g., using exact matching or designing test cases). | ★★★ | ☐ |
| 21 | Design appropriate evaluation metrics for the evaluation set (e.g., precision, accuracy, pass@K, recall). | ★★★ | ☐ |
| 22 | Consider the adequacy of the evaluation metrics (e.g., is the code coverage high enough). | ★★★ | ☐ |
| 23 | Consider if there are any other evaluation perspectives (e.g., readability, efficiency, safety, security). | ★ | ☐ |

*Figure 4.* Guideline for Benchmark Construction

| | Phase 3. Benchmark Evaluation | Priority | ☑ |
|---|---|---|---|
| 24 | Consider whether sufficient LLMs are evaluated. | ★ | ☐ |
| 25 | Consider whether representative LLMs (e.g., covering latest/classical LLM families, small/large LLMs, and open-/closed-source LLMs) are evaluated. | ★ | ☐ |
| 26 | Consider whether the prompt is of high quality (e.g., the instruction and intent are clear). | ★★★ | ☐ |
| 27 | The prompts have been validated by humans or LLMs (e.g., evaluated or discussed by participants or preliminarily tried out on several LLMs). | ★ | ☐ |
| 28 | Try different paraphrases of the prompt. | ★ | ☐ |
| 29 | Try different prompting strategies to observe the impact on the evaluation results (e.g., in-context learning, chain-of-thought). | ★★ | ☐ |
| 30 | Pay attention to the hardware environment (such as GPU card, storage size, etc.) of the experiment. | ★★★ | ☐ |
| 31 | Pay attention to the operating system and software environment (e.g. operating system, version, etc.) used for the experiment. | ★★★ | ☐ |
| 32 | Pay attention to the off-the-shelf platforms, frameworks, or libraries for LLM evaluation (e.g., fast chat, vllm, huggingface) that are used. | ★★ | ☐ |
| 33 | Repeat the experiment multiple times to reduce the impact of randomness on the evaluation. | ★ | ☐ |
| 34 | Consider various randomization strategies (e.g., trying various temperature parameters) to reduce the impact of parameter configuration on the evaluation. | ★★ | ☐ |
| 35 | Record the experimental process in detail (e.g., parameter settings, running time, input/output pairs, etc.). | ★★★ | ☐ |

*Figure 5.* Guideline for Benchmark Evaluation

| | Phase 4. Benchmark Analysis | Priority | ☑ |
|---|---|---|---|
| 36 | Observe the difficulty of the benchmark, checking if the benchmark is too hard or too easy for LLMs (i.e., most LLMs score too high/low). | ★★ | ☐ |
| 37 | Consider whether the benchmark can distinguish the pros and cons of different LLMs. | ★ | ☐ |
| 38 | If the experiment is repeated several times, consider the stability of the benchmark (i.e., whether the experimental results vary too much in the repeated experiments). | ★ | ☐ |
| 39 | Analyze the correlation between the data and their score. For example, if there is a correlation between the data (such as similar difficulty and knowledge required), then the scores should also be correlated. | ★★ | ☐ |
| 40 | Compare the performance of LLMs on this benchmark with their performance on **other related benchmarks**. | ★ | ☐ |
| 41 | Consider presenting the experiment results in an appropriate way (e.g., table, line graph, pie chart, etc.). | ★★★ | ☐ |
| 42 | Consider presenting the experiment results clearly (e.g., distinguishable colors/labels/shapes, etc.). | ★★★ | ☐ |
| 43 | Explain the experiment results. | ★★★ | ☐ |
| 44 | Observe correlations via multiple perspectives from the experimental results (e.g., performance is correlated with model size or amount of context). | ★ | ☐ |

*Figure 6.* **Guideline for Evaluation Analysis**

| | Phase 5. Benchmark Release | Priority | ☑ |
|---|---|---|---|
| 45 | The analysis of the evaluation results will be inspiring (e.g., shed light on future direction, make actionable advice, etc.). | ★ | ☐ |
| 46 | Set the appropriate license for the benchmark. | ★★★ | ☐ |
| 47 | Review the released benchmark or other artifacts to ensure they do NOT contain sensitive information (e.g., API keys, usernames, passwords, etc.). | ★★★ | ☐ |
| 48 | review the released benchmark or other artifacts to ensure they do NOT contain toxicity information (e.g., abusive comments/identifiers). | ★★★ | ☐ |
| 49 | Make sure the benchmark is open-accessible. | ★★★ | ☐ |
| 50 | Make sure the test cases or reference data are open and accessible. | ★★★ | ☐ |
| 51 | Provide prompts used in the experiment to ensure the experiments are reproducible. | ★★★ | ☐ |
| 52 | Disclose the experimental environment (e.g., hardware, operating system, software version, framework platform) to ensure the reproducibility of the experiment. | ★★★ | ☐ |
| 53 | Make the detailed experimental results public for verification. | ★★★ | ☐ |
| 54 | Ensure the quality of the user manual such as README (e.g., it contains necessary benchmark introduction, executable scripts, etc.). | ★★ | ☐ |
| 55 | Provide convenient evaluation interfaces for the released benchmark (e.g., providing a command line interface, docker, etc.). | ★★ | ☐ |

*Figure 7.* **Guideline for Benchmark Release**

# C. Statistics of studied benchmarks

In this section, we conducted a comprehensive and detailed statistical analysis of the 572 benchmarks collected.

## C.1. Profile of Studied Benchmarks

We first show the trend in the development of benchmarks from 2014 to 2025. As shown in Figure 8, the data shows a modest beginning, with only a handful of benchmarks created annually until 2017. From 2018 onwards, there has been a noticeable uptrend in benchmark creation, culminating in a significant jump to 210 benchmarks in 2024, and 316 benchmarks in 2025. This sharp increase indicates a recent heightened interest and demand for comprehensive code-related benchmarks for LLMs, reflecting the evolving complexities and expanding requirements of automated software engineering.

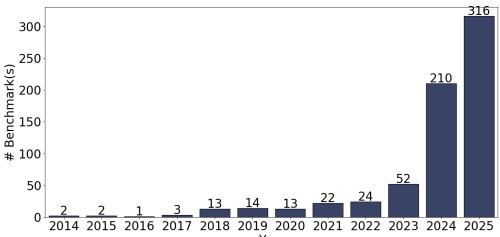

*Figure 8.* Benchmark Distribution over Years

**Hierarchy of Benchmarks**. Figure 59 visualizes the inheritance relationships among benchmarks, indicating that the benchmarks on the *left serve as sources* for those on the right. It highlights that *18% of benchmarks act as data sources*, continuously benefiting the construction of subsequent benchmarks.

Figure 59 reveals that ***HumanEval*** (Chen et al., 2021a), as the ***most significant source*** benchmark, benefits at least *15 downstream benchmarks*, followed by MBPP (Austin et al., 2021) and CodeSearchNet (Husain et al., 2019). From the right side of the figure, some benchmarks, like VulBench (Gao et al., 2023b), incorporate methodologies or data from 4 previous benchmarks, and codeRag-Bench (Wang et al., 2024f) integrates elements from 8 prior benchmarks.

This hierarchical structure among benchmarks also alerts us that the ***data quality of a benchmark not only affects its own credibility but can continue to impact others*** if it serves as a source. This underscores the importance of adhering to stringent guidelines during benchmark development and highlights the crucial role of ***establishing standards*** to ensure the integrity and utility of benchmark data across research and development efforts.

***Coding Task.*** Regarding the ***coding tasks***, Figure 9 illustrates the distribution of various coding tasks across bench-

marks. It is clear that the task of ***Code Generation*** is most prevalent, with 235 benchmarks focusing on this area, according to 34.97% (235/672) of studied benchmarks, indicating a significant interest in generating code automatically. The second most prevalent is code reasoning 12.64% (85/672), followed by Program Repair and Defect Detection.

When examining the distribution of coding tasks by year (see Figure 10), we observe that benchmark growth accelerated most rapidly in the past two years (2024–2025), with code generation remaining the most frequently benchmarked task. At the same time, demand for code reasoning has increased substantially, rising from 19 benchmarks in 2024 to 61 in 2025. Benchmarks for program repair have also nearly doubled over the same period, increasing from 22 to 33.

These trends suggest a growing emphasis on LLMs' reasoning capabilities as well as their role in code development and software maintenance, reflecting an evolving expectation of model competence beyond surface-level code synthesis.

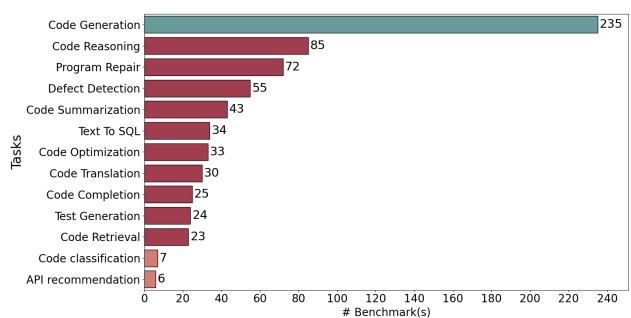

*Figure 9.* Benchmark Distribution over Tasks

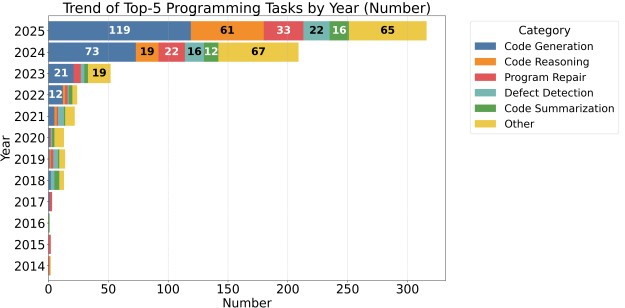

*Figure 10.* Benchmark Distribution over Tasks per Year

***Programming Languages.*** Figure 11 shows the distribution of benchmarks across various programming languages. The overall trend indicates a strong preference for benchmarking ***Python***, which leads with 409 (71.50%) benchmarks, followed by Java and C++, with 229 and 160, respectively. The graph also reveals a diverse range of languages being used. In total, 67 programming languages are studied by

these 672 benchmarks. Though some programming languages, such as Kotlin, Swift, and Scala, are less frequently exercised, the benchmarks involving them are tailored to different application needs and technology environments. This distribution shows the existing benchmarks are dominated by three mainstream programming languages, leaving other programming languages less studied and benchmarked.

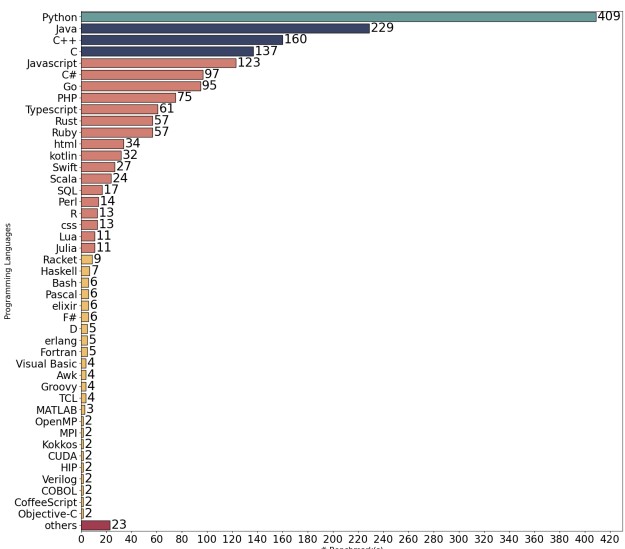

*Figure 11.* Benchmark Distribution over Programming Language

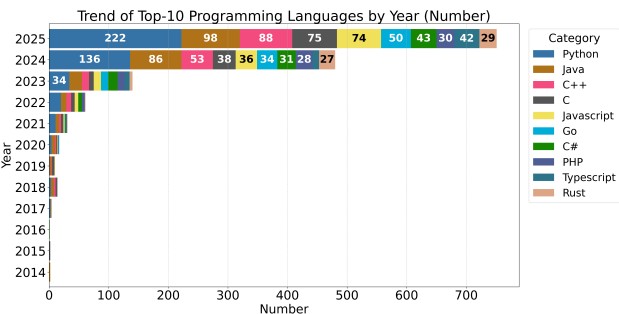

*Figure 12.* Benchmark Distribution over Programming Language per Year

When analyzed by year (Figure 12), the number of benchmarks for C/C++ and JavaScript has increased noticeably between 2024 and 2025. We also observe a sharp rise in Rust-related benchmarks: although only a few benchmarks existed for Rust since its release in 2015, 57 benchmarks were published in the past two years (27 in 2024 and 29 in 2025). These trends in programming language coverage reflect the evolving demand for LLMs across different languages, highlighting particularly the growing reliance on large models to generate, reason about, and maintain code in these languages.

*Natural Language*. Figure 13 illustrates the distribution

of benchmarks for different natural languages. The bar chart overwhelmingly shows that English is the dominant language, with 577 (85.9%) benchmarks highlighting its ubiquity in research and development. Other languages have significantly fewer benchmarks, with 16 in Chinese and 15 in Russian. The category labeled "Other" includes 24 benchmarks spread across other natural languages, indicating some diversity but limited attention to non-English benchmarks. This distribution highlights the prominence of English in the global research community and also demonstrates the ***uneven representation*** of natural languages in the studied benchmarks.

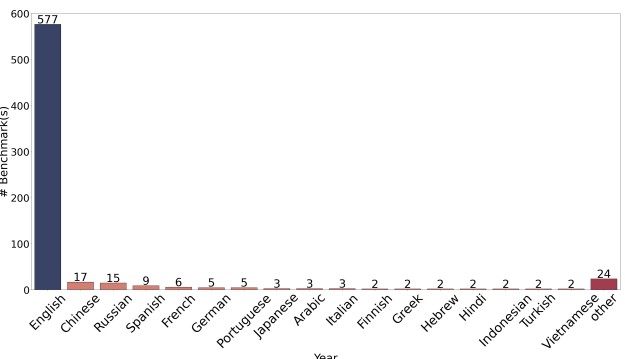

*Figure 13.* Benchmark Distribution over Natural Language

***Modals in the benchmarks***. Figure 14 presents the distribution of benchmarks according to the type of language used in their prompts. The chart shows that the majority, at 66.2%, of the benchmarks use a combination of natural language and programming Language, followed by PL only (13.7%) and NL only (18.2%).

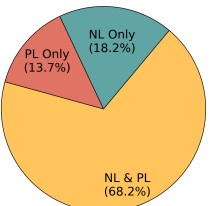

*Figure 14.* Benchmark Distribution over Modal in Prompt

***Granularity***. The code snippet in a code-related benchmark varies from statement-level (i.e., one line of code. For example, CoNaLa (Yin et al., 2018) and Math-QA (Amini et al., 2019)), function-level (i.e., a function unit of code. For example, HumanEval (Chen et al., 2021a) and MBPP (Austin et al., 2021)), class-level (i.e., a class with multiple function units of code. For example, ClassEval (Du et al., 2023)) and project-level (i.e., a project with multiple classes or modules. For example, DevEval (Li et al., 2024a) and JavaBench (Cao et al., 2024a)).

Figure 15 illustrates the granularity levels at which bench-

marks are typically conducted. The chart shows that the majority of benchmarks, comprising **66.9%, focus on the function level**. Projects constitute 21.5% of the benchmarks. Class-level granularity is the least represented at only 3.4%.

When analyzed by year, an interesting phenomenon emerges (Figure 16): between 2024 and 2025, the number of project-level benchmarks surged, increasing from 35 to 95 new benchmarks. Importantly, these numbers refer to benchmarks newly introduced in each year, rather than cumulative totals. This surge indicates a growing community focus on the real-world applicability and large-scale practical utility of LLMs.

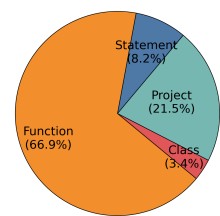

*Figure 15.* Benchmark Distribution over Granularity

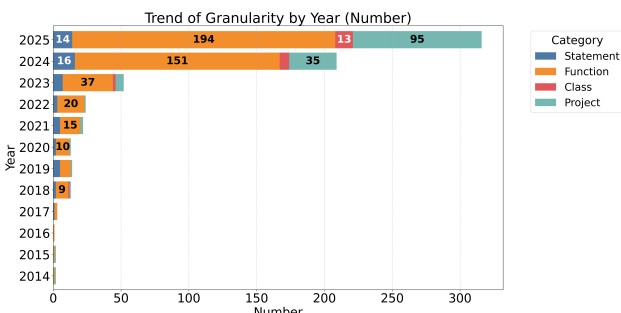

*Figure 16.* Benchmark Distribution over Granularity per Year

### C.2. Statistics about Benchmark Design

***Design of Studied Capabilities***. To understand whether benchmark developers recognize the capabilities of LLMs they aim to evaluate, we carefully analyzed 30 representative benchmarks (Appendix E) to see if they clearly specify the capabilities being assessed by their benchmarks. As shown in Figure 17, 80% of benchmarks explicitly specify the capabilities (e.g., intention understanding, problem solving, testing, debugging capabilities)to be evaluated, while 20% do not. The statistics show that the most highly cited benchmarks clearly define the assessment capabilities.

Furthermore, we investigated the 30 focused benchmarks and identified a case (Figure 18) from MBPP (Austin et al., 2021) where the case is likely to fall outside of the targeted capability of the benchmark. In particular, MBPP (Austin et al., 2021) aims to "*measure the ability of these models to synthesize short Python programs from natural language*

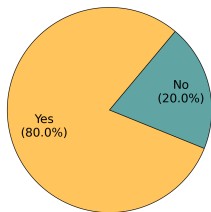

*Figure 17.* Benchmark Distribution Over Capabilities Consideration

*descriptions*" for "*entry-level programmers*". As we can see from Figure 18, the prompt requires LLMs to "*Write a function to calculate the dogs' years.*" Simply from this description, an entry-level programmer is unlikely to write a correct program without knowing the conversion equation from dogs' year to dogs' age. In other words, this case is more about assessing whether LLMs have acquired this specific knowledge rather than evaluating the most fundamental programming skills.

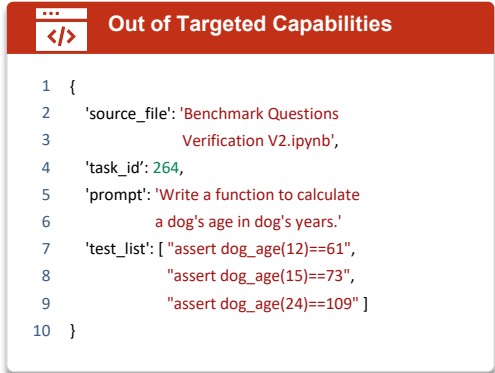

*Figure 18.* An Example of Out-of-capability Case from MBPP (Austin et al., 2021).

***Design of Studied Application Scenarios***. Similarly, to understand whether benchmark developers scoped the application scenarios of LLMs they aim to evaluate, we carefully analyzed 30 representative benchmarks (Appendix E) to see whether they explicitly specify the application scenarios their benchmarks target. As shown in Figure 19, 76.7% representative benchmarks have clearly specified application scenarios (e.g., programming assistant), while the rest do not. Indeed, clearly defining the application scenarios could help benchmark constructors establish precise goals for the design and development of the benchmark, ensuring accuracy in the evaluation.

### C.3. Statistics about Data Preparation

#### C.3.1. DATA PREPROCESSING

***Data Deduplication.*** During benchmark preparation, data cleaning and preprocessing are necessary. However, as shown in Figure 20, only **32.9% benchmarks have dedu-**

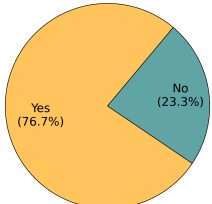

*Figure 19.* Benchmark Distribution Over Expected Application Scenario Consideration

**plicated** the collected data. More than half of them didn't mention this process.

The yearly trend (Figure 21) shows that, although the number of benchmarks that applied deduplication increased slightly from 71 in 2024 to 93 in 2025, the total number of benchmarks in 2025 more than doubled compared to 2024. As a result, the absolute number of benchmarks that ignored or did not apply deduplication nearly doubled, highlighting a growing risk of duplicated data despite increased awareness.

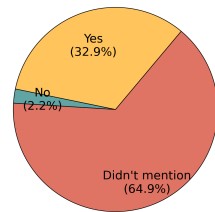

*Figure 20.* Benchmark Distribution over Deduplication

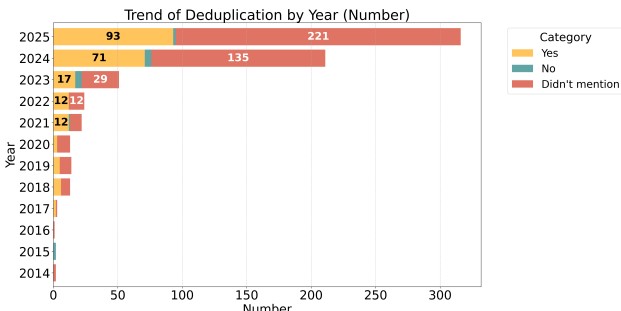

*Figure 21.* Benchmark Distribution over Deduplication per Year

To investigate the situation, we went through the 30 representative benchmarks (Listed in Appendix E) and found two duplicated subjects in MBPP (Austin et al., 2021). Tasks with id `71` and `141` examined the same functionality, i.e., "*Write a function to sort a list of elements*.", collected from the same source.

***Data Quality Assurance.*** Ensuring data quality for the benchmark is essential. However, our statistics (Figure 23) show disappointing results. ***46.0% of benchmarks do not take any measures for data quality assurance***. Among

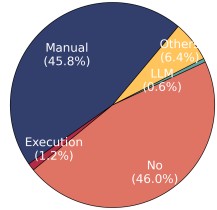

*Figure 22.* A Counterexample of Rule 16 from MBPP (Austin et al., 2021).

those benchmarks that do incorporate data quality measures, the majority rely on manual checks, which account for 45.8%. Other countermeasurements, such as code execution, constitute only 1.2%. Additional methods, such as using download counts as a basis, are also employed (6.4%) .

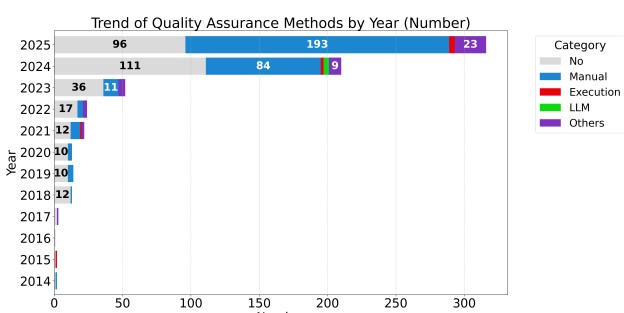

*Figure 23.* Benchmark Distribution over Quality Assurance Method

*Figure 24.* Benchmark Distribution over Quality Assurance Method per Year

Examining the trends by year (Figure 24), **one positive development is the increasing prevalence of manual quality checks**. The number of benchmarks with manual check guarantees doubled from 84 in 2024 to 193 in 2025, indicating that a growing share of code benchmarks now undergo partial or full human verification.

Additionally, we dived into the 30 representative benchmarks (Listed in Appendix E) and identified an example

where the code cannot be executed successfully. As shown in Figure 25, the function `swap()` in line 7 has not been defined, so the execution of the code would fail if the code has been executed. This highlights a significant gap in ensuring the reliability and validity of benchmark data, underscoring the need for more rigorous and automated data quality assurance practices.

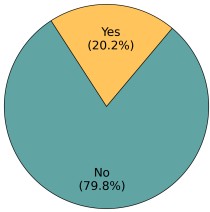

```
1  Import math
2  def  min_Operations (A, B):
3      """ Write a python function to find
4      the minimum operations required
5      to make two numbers equal. """
6      if (A > B):
7          swap(A,B)
8      B = B // math.gcd(A,B);
9      return B - 1
```

*Figure 25.* An Example from MBPP (Austin et al., 2021) that failed to be executed.

***Data Contamination Resolution.*** Data contamination (Golchin & Surdeanu, 2023; Cao et al., 2024b) threat has been widely discussed. A benchmark with contaminated data may yield overclaimed results, misleading the understanding of the LLMs' capabilities. According to our statistics (Figure 26), most (79.8%) benchmarks were not aware of and have not taken any measures to alleviate data contamination, being vulnerable to data contamination threats. The yearly trend (Figure 27) further highlights the persistent neglect of data contamination in benchmark construction. From 2024 to 2025, the number of benchmarks that did not account for data contamination nearly doubled, increasing from 165 to 236, indicating that this risk persists or has become more widespread.

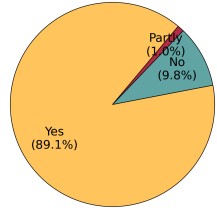

*Figure 26.* Benchmark Distribution over Quality Assurance on Data Contamination

### C.3.2. STATISTICS ABOUT DATA CURATION

***Ground truth solutions***. Figure 28 shows that although the majority (89.1%) of benchmarks provide reference code as ground truth, there are 9.8% of benchmarks without reference code. Although it is not compulsory as long as object measurements (e.g., test cases) are provided, ***a reference code is still recommended.*** Indeed, if a benchmark provides reference code, its reliability tends to be better because it en-

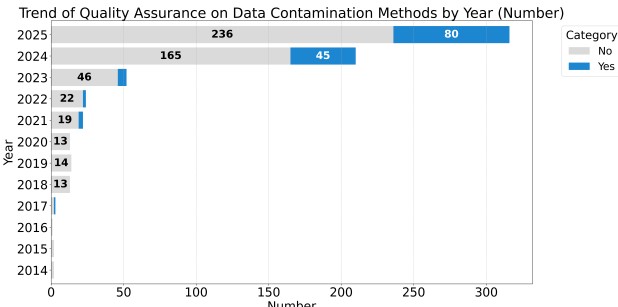

*Figure 27.* Benchmark Distribution over Quality Assurance on Data Contamination per Year

sures that there are feasible solutions for the tasks involved. This guarantees that the tasks are theoretically and practically solvable, enhancing the benchmark's usefulness and credibility in real-world applications.

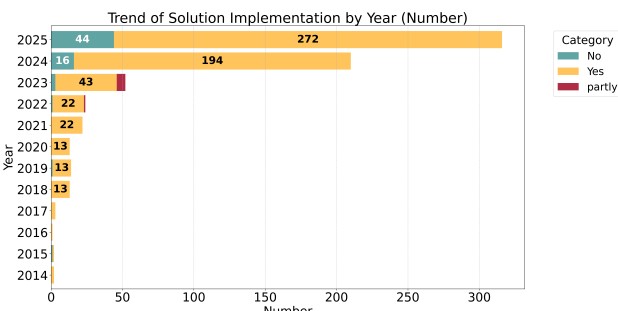

*Figure 28.* Benchmark Distribution over Solution

*Figure 29.* Benchmark Distribution over Solution per Year

From the year-by-year trend (Figure 29), the number of benchmarks without ground truth tripled in 2025, reaching 44 benchmarks. This may reflect **a shift toward more open-ended and challenging benchmarks**, where standard solutions do not yet exist or remain to be explored, highlighting the evolving nature of tasks designed for LLM evaluation.

Additionally, ***the correctness of the ground truth solution*** should also be noted. Figure 30 shows an ***incorrect*** code solution provided in HumanEval (Chen et al., 2021a). This should draw benchmark constructors' attention to the correctness of the benchmark reference code.

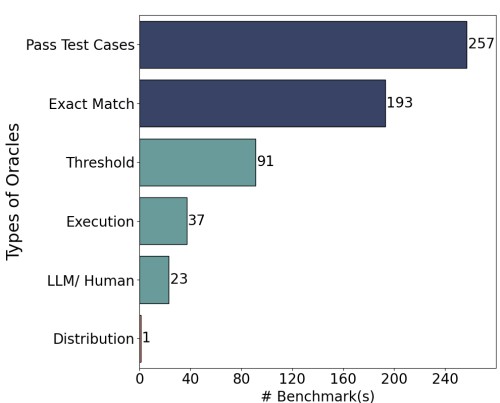

```
    Incorrect Ground-truth
1  def check_dict_case(dict):
2      """ Given a dictionary, return True if all keys are strings
3      in lower case or all keys are strings in upper case, else
4      return False. The function should return False is the
5      given dictionary is empty. """
6      if len(dict.keys()) == 0:
7          return False
8      else:
9          state = "start"
10         for key in dict.keys():
11             if isinstance(key, str) == False:
12                 state = "mixed"
13                 break
14             if state == "start":
15                 if key.isupper():
16                     state = "upper"
17                 elif key.islower():
18                     state = "lower"
19                 else:
20                     break
21             elif (state == "upper" and not key.isupper())
22                     or (state == "lower" and not
23                     key.islower()):
24                 state = "mixed"
25                 break
26             else:
27                 break
28         return state == "upper" or state == "lower"
```

*Figure 30.* An Example from HumanEval  (Chen et al., 2021a) which shows ***an incorrect solution*** provided in the benchmark.

***Oracle***. An oracle (Barr et al., 2014) is a way to determine whether the output is correct or not. For example, assume the output of LLMs is in the form of code, then an oracle could be running tests against the code and see whether the code can pass all the tests. Figure 31 shows the distribution of types of oracle that are used in these benchmarks. Clear that passing test cases (257 / 672 = 38.2%) and the exact match (193 / 672 = 28.7%) are the most common oracles used in code benchmarks, followed by thresholds (i.e., similarities smaller than a specific threshold).

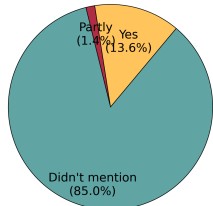

*Figure 31.* Benchmark Distribution over Test Oracle

***Code coverage*** (Ivanković et al., 2019), as a common oracle for code-related benchmarks, measuring the ratio of covered code by tests, has been widely adopted to determine the output correctness. It should be considered whether a benchmark uses test case passing as a criterion for the cor-

rectness of the generated code. Otherwise, a test could be too weak to detect the existence of a defect in the generated code. For example, as pointed out by prior work (Liu et al., 2023b), existing benchmarks such as HumanEval (Chen et al., 2021a) and MBPP (Austin et al., 2021) still suffer from "insufficient tests", allowing incorrect code to pass all the tests without capturing the bugs.

Despite its importance, as shown in Figure 32, among the benchmarks that use test cases as the oracle, only 13.6% considered and reported "test coverage" explicitly in their papers, while 85.0% ignored the test coverage.

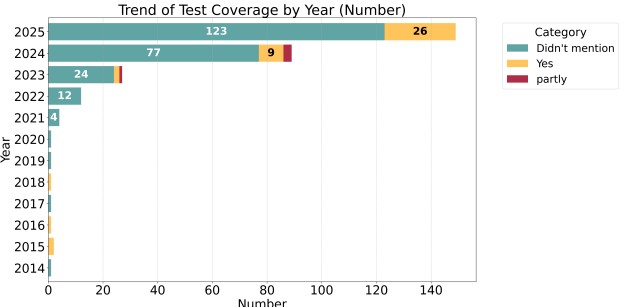

*Figure 32.* Benchmark Distribution over Test Coverage

*Figure 33.* Benchmark Distribution over Test Coverage per Year

The annual distribution (Figure 33) makes this trend even clearer: although many benchmarks in the past three years (2023–2025) did not consider code coverage, the sheer volume of benchmarks in 2025 means that the absolute number of such benchmarks is high, with 24, 77, and 123 benchmarks respectively for 2023, 2024, and 2025. This underscores that, despite growing awareness of evaluation rigor, a substantial number of benchmarks continue to provide incomplete testing, highlighting a persistent threat to the validity and reliability of benchmark-driven assessment.

Furthermore, we dived into 30 representative benchmarks (Listed in Appendix E) and identified an example (Figure 34) from MBPP (Austin et al., 2021) where the test is incorrect. It alerts us that both the quality of the test and the test adequacy (e.g., code coverage) should be considered.

### C.4. Statistics about Evaluation

***Studied LLMs***. We summarize the number of LLMs that have been evaluated in each benchmark evaluation. Among

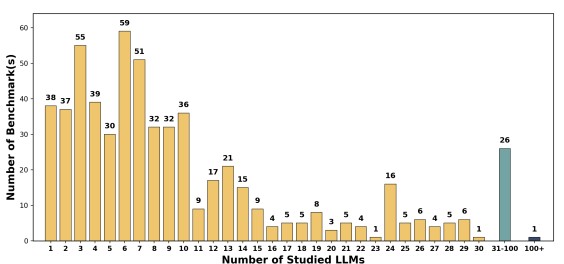

```
Wrong Example Tests
1  {
2    'source_file': 'charlessutton@: Benchmark
3              Questions Verification V2.ipynb',
4    'task_id': 461,
5    'prompt': 'Write a python function to count the
6              upper case characters in a given string.'
7    'test_list': [ "assert upper_ctr('PYthon') == 1",
8              "assert upper_ctr('BigData') == 1",
9              "assert upper_ctr('program') == 0" ]
10 }
```

*Figure 34.* An Example of Incorrect Tests from MBPP (Austin et al., 2021).

the 672 benchmarks, 585 of them are evaluated over LLMs, so we show the statistics over them. As shown in Figure 35, most benchmarks were evaluated against six LLMs (10.0% = 59 / 585), followed by three LLMs.

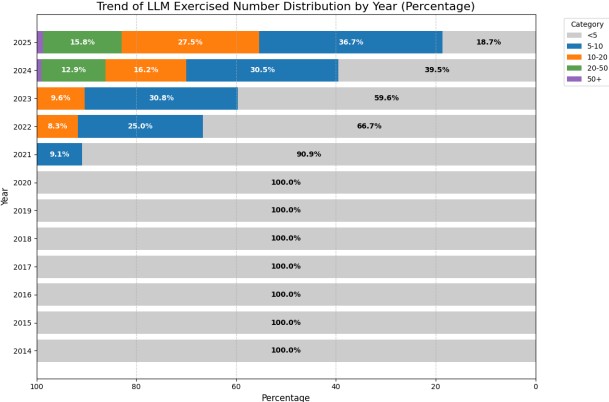

*Figure 35.* Benchmark Distribution over LLM Experimented

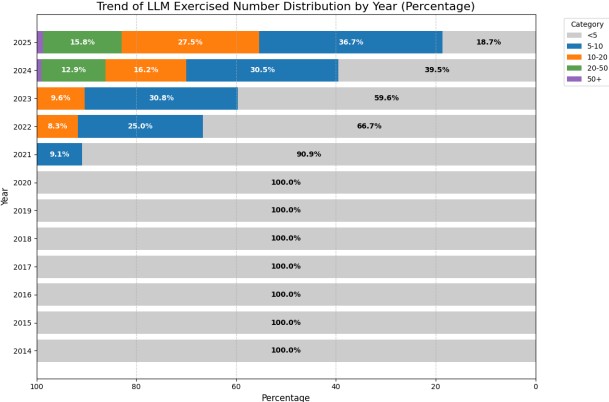

*Figure 36.* Benchmark Distribution over LLM Experimented Per Year

A positive trend also emerges when examining the year-by-year distribution (Figure 36). In 2024, 39.5% of benchmarks were evaluated against fewer than five LLMs, whereas in 2025, 64.2% of benchmarks (36.7% + 27.5%) were evaluated against 5–20 models. This shift indicates that benchmark studies are increasingly aiming for more comprehensive and generalizable evaluations. However, it also in-

troduces substantially higher computational and financial costs, highlighting the trade-off between evaluation rigor and resource efficiency.

Additionally, we listed the top-10 LLMs by the number of code-related benchmarks they have been evaluated, as shown in Figure 37. GPT series leads significantly with 446 benchmarks, suggesting its widespread adoption and possibly its versatility or superior performance in handling code-related tasks. The rest, including DeepSeek, Qwen, and others, show varying degrees of involvement, with numbers ranging from 269 down to 64 benchmarks for Claude. This figure may provide a reference for choosing which model to evaluate. In addition, it is worth mentioning that different LLMs should be considered for different coding tasks.

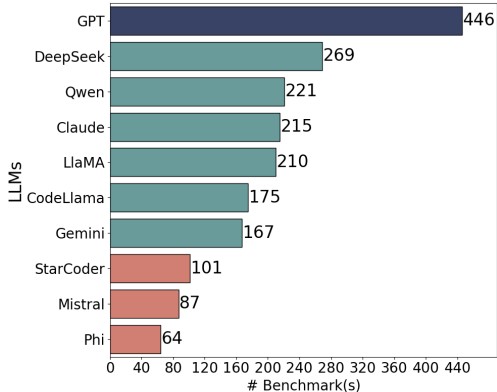

*Figure 37.* Top-10 Studied LLMs for Code-related Benchmarks

***Experiment Environments.*** The experimental environment (such as the operating system and hardware) is important for the reproduction of the experiment. However, Figure 39 and Figure 38 highlight a significant gap. Only 24.6% of benchmarks document the devices used in their experiments, leaving a substantial 75.4% that do not. The situation appears even more dire when considering os, with only 6.3% of benchmarks documenting the OS used, while a staggering 93.7% neglect to record this information.

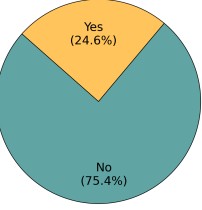

*Figure 38.* Benchmark Distribution over Recording Experiment Devices

***Prompting and Prompting Strategies*** Prompting has a direct impact on the quality of the LLMs' output results (Wei et al., 2022; He et al., 2024a; Jin et al., 2024). So, we summarized whether different prompting strategies have been

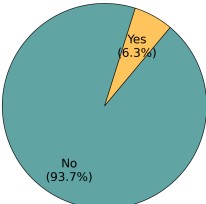

*Figure 39.* Benchmark Distribution over Recording Experiment OS

evaluated and statistics the distribution. Figure 40 shows the usage of four kinds of prompts: zero-shot, few-shot, chain-of-thought, and retrievals (RAG). From Figure 40, we can see that a vast majority (89.4%) of benchmarks were evaluated in a zero-shot context setting, while only 18.6% of benchmarks were evaluated in a few-shot manner. Even fewer benchmarks were evaluated under the Chain-Of-Thought (CoT) and RAG settings, utilized by only 3.7% and 1.0%.

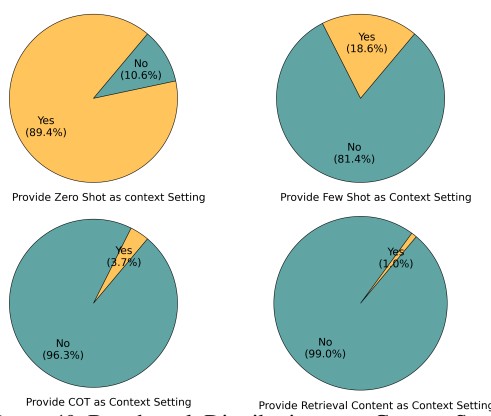

*Figure 40.* Benchmark Distribution over Context Setting

***Prompt Quality*** The prompt quality also greatly impacts the LLM evaluation (He et al., 2024b). So, carefully designing a prompt needs consideration. However, as shown in Figure 41, 76.7% representative benchmarks (Appendix E) do not validate whether the prompt they used is well-designed.

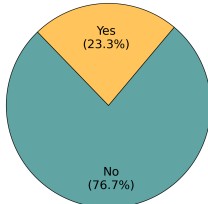

*Figure 41.* Benchmark Distribution Over Validation of Prompts

***Repeated Experiment*** Given the random nature of LLMs, the experiments are expected to repeat, ensuring the stability and reliability of the results. However, as shown in Figure 42, ***only 33.4% benchmarks went through a re-***

***peated experiment***, while a majority of 66.6% opted against repeating the experiment. This reflects a lack of awareness regarding the stability and reproducibility of evaluations.

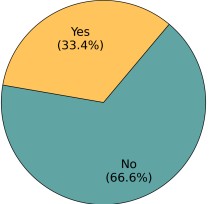

*Figure 42.* Benchmark Distribution over Repeating the Experiment

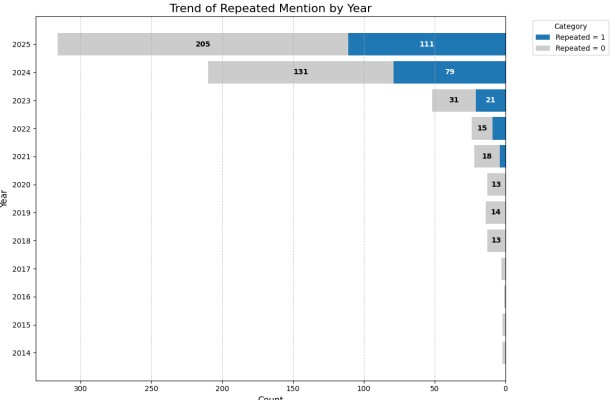

*Figure 43.* Benchmark Distribution over Repeating Count the Experiment Per Year

Examining trends year by year (Figure 43), the number of benchmark developers who repeated experiments increased in 2025 compared to 2023 and 2024 (31 in 2023, 79 in 2024, and 111 in 2025). However, because the total number of benchmarks grew substantially in 2024 and 2025, the absolute number of benchmarks without repeated experiments remains the highest, totaling 205 benchmarks, which accounts for 30.5% (205/672) of all benchmarks. This indicates that, despite increasing awareness of reproducibility, a significant fraction of benchmarks still lack repeated verification, posing a persistent risk to evaluation reliability.

### C.5. Statistics about Analysis

***Experiment Explanation***. Explaining experiment results is crucial for other practitioners to understand what the outcomes mean in the context of the research questions. So, we investigate whether the representative benchmarks (Appendix E) have explained the experiment results. As shown in Figure 44, 63.3% benchmarks have detailed explanations and analyses of their evaluation results, while still 36.7% have not. Indeed, an explanation contributes to the body of knowledge by making it possible to understand and compare results with previous studies, promoting transparency within the community.

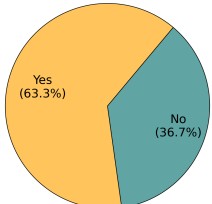

*Figure 44.* Benchmark Distribution Over Explaining the Experiment

A clear and precise presentation of experimental results is important for enabling robust interpretation and comparison across benchmarks. However, further examination of the 30 representative benchmarks (listed in Appendix E) revealed notable deficiencies in result visualization. As shown in Figure 45, CruxEval (Gu et al., 2024) exhibits unclear experimental result presentation. Specifically, the scatter plot suffers from ambiguous labeling, poor readability of axis values, and inconsistent marker encoding, making it difficult for researchers to extract meaningful insights. Such presentation shortcomings obscure the performance relationships between methods and compromise the benchmark's usability for fair evaluation. To address these issues, benchmarks should adopt standardized and well-documented visualization practices, ensuring results are interpretable, accessible, and reproducible.

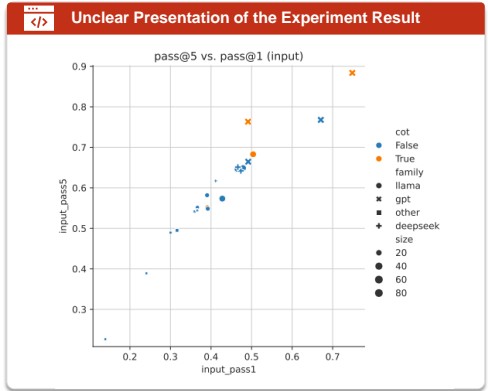

*Figure 45.* An Example of Unclear Experiment Analysis and Display from CruxEval (Gu et al., 2024)

### C.6. Statistics about Release

***Data Accessibility***. The fundamental requirement for releasing a benchmark is that it must be open-sourced. However, surprisingly, as shown in Figure 46, we observed that 2.2% of the benchmarks are only partially open-sourced (e.g., missing some subjects or tests), and ***14.7% are not open-sourced at all*** (e.g., links/web pages are no longer active).

Fortunately, the year-by-year trend (Figure 47) shows a positive shift toward openness. From 2024 to 2025, the number of open-sourced benchmarks increased substantially, rising

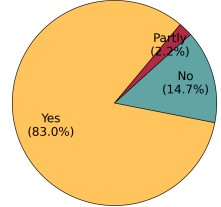

*Figure 46.* Benchmark Data Availability

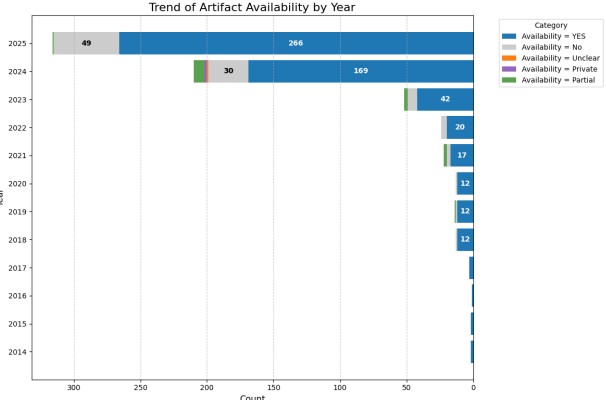

*Figure 47.* Benchmark Data Availability Count Per Year

from 169 to 266, indicating growing community commitment to transparency, accessibility, and reproducibility.

***Prompt Accessibility***. Detailed prompts are essential for ensuring the reproducibility and transparency of code-related benchmarks. Yet, Figure 48 indicates that ***38.2% of benchmarks do not provide detailed prompts***, limiting the ability to accurately replicate and evaluate the performance of LLMs. Lack of prompt disclosure highlights a gap in benchmark design practices, as prompts are often indispensable for understanding model performance under specific conditions, raising concerns about the consistency and reproducibility of reported results.

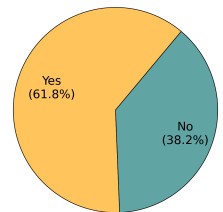

*Figure 48.* Availability of Prompts

When examined on a year-by-year basis Figure 49, from 2024 to 2025, although the number of benchmarks that release prompts increased from 140 to 233, the number of benchmarks that do not release prompts also grew, from 64 to 82. This indicates that transparency has improved in absolute terms, but has not kept pace with the rapid growth

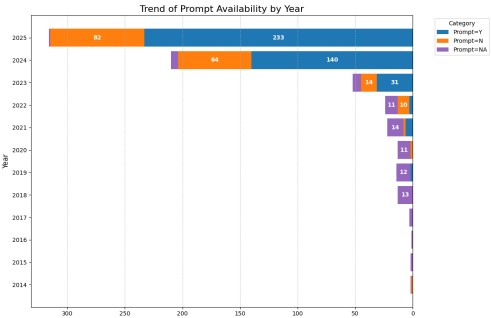

*Figure 49.* Availability of Prompts Per Year

that provide an overview of the benchmark, step-by-step execution guides, and troubleshooting instructions, ensuring a seamless and reproducible user experience.

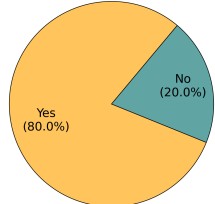

*Figure 51.* Availability of User Manual

in benchmark releases.

***Logging Info Accessibility***. Providing detailed logging information, including comprehensive experimental results, is essential for ensuring transparency, verifiability, and reproducibility in benchmarking research. However, as shown in Figure 50, only ***16.7% of the benchmarks make their experimental results publicly available***, while 80.0% fail to disclose this critical information. Alarmingly, an additional 3.3% provide only partial logging details, further complicating result verification. The absence of complete logging information creates significant barriers for researchers attempting to reproduce experiments or validate reported findings, thereby undermining the reliability of benchmarks. To address this, we emphasize the necessity of making detailed logging information, including intermediate results and metrics, publicly accessible to uphold rigorous scientific standards and foster trustworthy comparisons across models.

***Convenient Evaluation Interface Availability***.

Providing convenient evaluation interfaces is essential for enhancing the usability and accessibility of benchmarks, enabling researchers to easily reproduce results and compare models. As shown in Figure 52, ***20% of benchmarks fail to offer any evaluation interfaces***, imposing significant barriers to usability. While a majority of benchmarks (80%) provide such interfaces, including command-line tools, Docker images, or scripts, the absence of standardized and user-friendly evaluation tools in a notable minority of cases highlights an area for improvement. Benchmarks without convenient evaluation interfaces require users to spend additional effort in setup and result verification, which can discourage adoption and hinder reproducibility. To address this, we emphasize the importance of releasing benchmarks with well-documented, ready-to-use evaluation pipelines to promote efficient, reliable, and fair benchmarking practices.

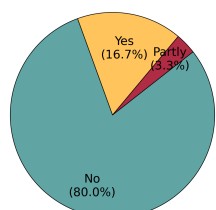

*Figure 50.* Availability of Logging Information

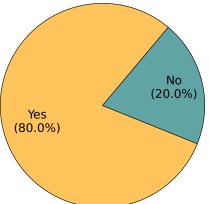

*Figure 52.* Availability of Convenient Evaluation Interfaces

***User Manual Accessibility***. A high-quality user manual, such as a well-documented README file, is crucial for enhancing benchmark usability, enabling users to understand the dataset, execute provided scripts, and reproduce results efficiently. However, our analysis revealed that a significant number of benchmarks lack comprehensive user manuals, hindering accessibility and adoption. As depicted in Figure 51, poorly structured or incomplete manuals often omit essential components such as benchmark introductions, usage instructions, and evaluation scripts. This creates unnecessary barriers for researchers who rely on these manuals for setup and experimentation. To address this, we advocate for benchmarks to include clear, standardized user manuals

***Temperature Records***. One critical parameter for benchmarking is the temperature setting, which influences stochasticity in LLMs. As shown in Figure 53, we observed that ***49.5% of benchmarks fail to record the temperature setting***, hindering reproducibility and fair evaluation. While 50.5% of benchmarks do document this parameter, the majority omission highlights an overlooked yet essential aspect of benchmark transparency.

***License Provision***. Releasing benchmarks under a clear and accessible license is fundamental for legal compliance and ensuring open collaboration. Figure 54 reveals that ***19.3% of benchmarks do not provide a license***, limiting their usability and distribution. Encouragingly, 80.7% of

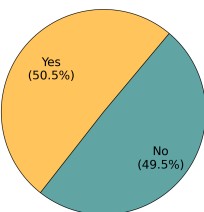

*Figure 53.* Benchmark Distribution over Recording Temperature

benchmarks do include a license, but the lack of licensing in nearly one-fifth of the benchmarks raises concerns about widespread adoption and usage. These findings emphasize the need for standardized practices in benchmark releases to promote legal clarity and accessibility.

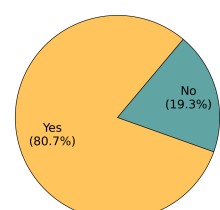

*Figure 54.* Provision of License

***Data Security***. Ensuring data security is a critical yet often overlooked aspect of benchmark development. Sensitive information, such as API keys, credentials, or private tokens, should never be included in benchmark releases. However, further investigation into 30 representative benchmarks (listed in Appendix E) revealed instances of sensitive data leakage. As shown in Figure 55, XSemPLR (Zhang et al., 2023b) inadvertently included an ***API key*** in its release, a critical oversight that can expose resources to external exploitation. Similarly, Figure 56 highlights an example from

```
API Key Leakage
1   #! /bin/bash
2
3   # conda activate skg
4   #export WANDB_API_KEY=*************************
5   export WANDB_PROJECT=mt5-large_mgeoquery_few-shot
6   export CUDA_LAUNCH_BLOCKING=1
```

*Figure 55.* An Example of API Key Leakage in Benchmark Release from XSemPLR (Zhang et al., 2023b).

CrossVul (Nikitopoulos et al., 2021), where ***personal names and email addresses*** were unintentionally disclosed. Such leakage poses risks of unauthorized access and resource misuse, potentially compromising systems and research integrity.

***Usability***. Clear and comprehensive documentation is crucial for ensuring the usability of benchmarks, as poorly

```
Name or Email Leakage
10942   Individual \fIreadline\fP initialization file
10943   .PD
10944   .SH AUTHORS
10945   Brian Fox, Free Software Foundation
10946   .br
10947   bfox@gnu.org
10948   .PP
10949   Chet Ramey, Case Western Reserve University
10950   .br
10951   chet.ramey@case.edu
```

*Figure 56.* An Example of Name & Email Leakage in Benchmark Release from CrossVul (Nikitopoulos et al., 2021).

written instructions can significantly hinder adoption and reproducibility. We dived into the 30 representative benchmarks (listed in Appendix E) and identified an example where the README file provided insufficient and unclear information. As shown in Figure 57, VulDeePecker (Li et al., 2018b) includes a less-than-ideal ReadMe file, which lacks essential usage instructions and evaluation guidelines, making the benchmark difficult to understand and deploy.

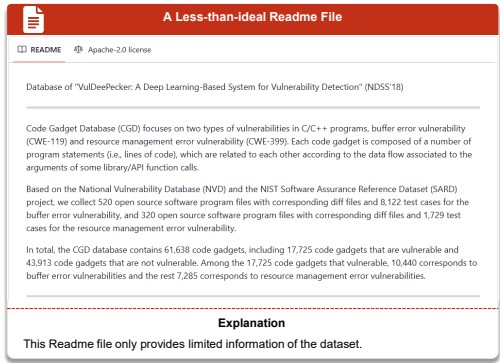

*Figure 57.* An Example of Unreadable and Hard-to-Use README in Benchmark Release from VulDeePecker (Li et al., 2018b).

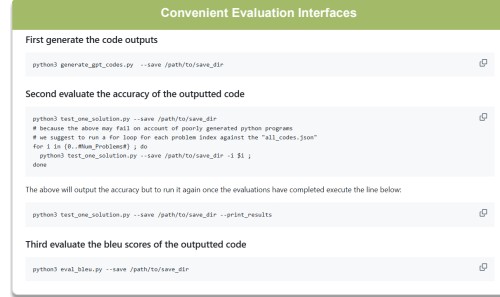

*Figure 58.* A Good Example of Easy-to-Read README in Benchmark Release from APPS (Hendrycks et al., 2021).

In contrast, Figure 58 highlights APPS (Hendrycks et al., 2021), which provides well-structured and easy-to-follow documentation. The APPS benchmark includes step-by-step instructions for generating, evaluating, and analyzing results, enabling users to efficiently reproduce experiments.

These observations emphasize the importance of high-

quality documentation for benchmarks to enhance acces-
sibility, reduce friction in usage, and foster reproducible
research.

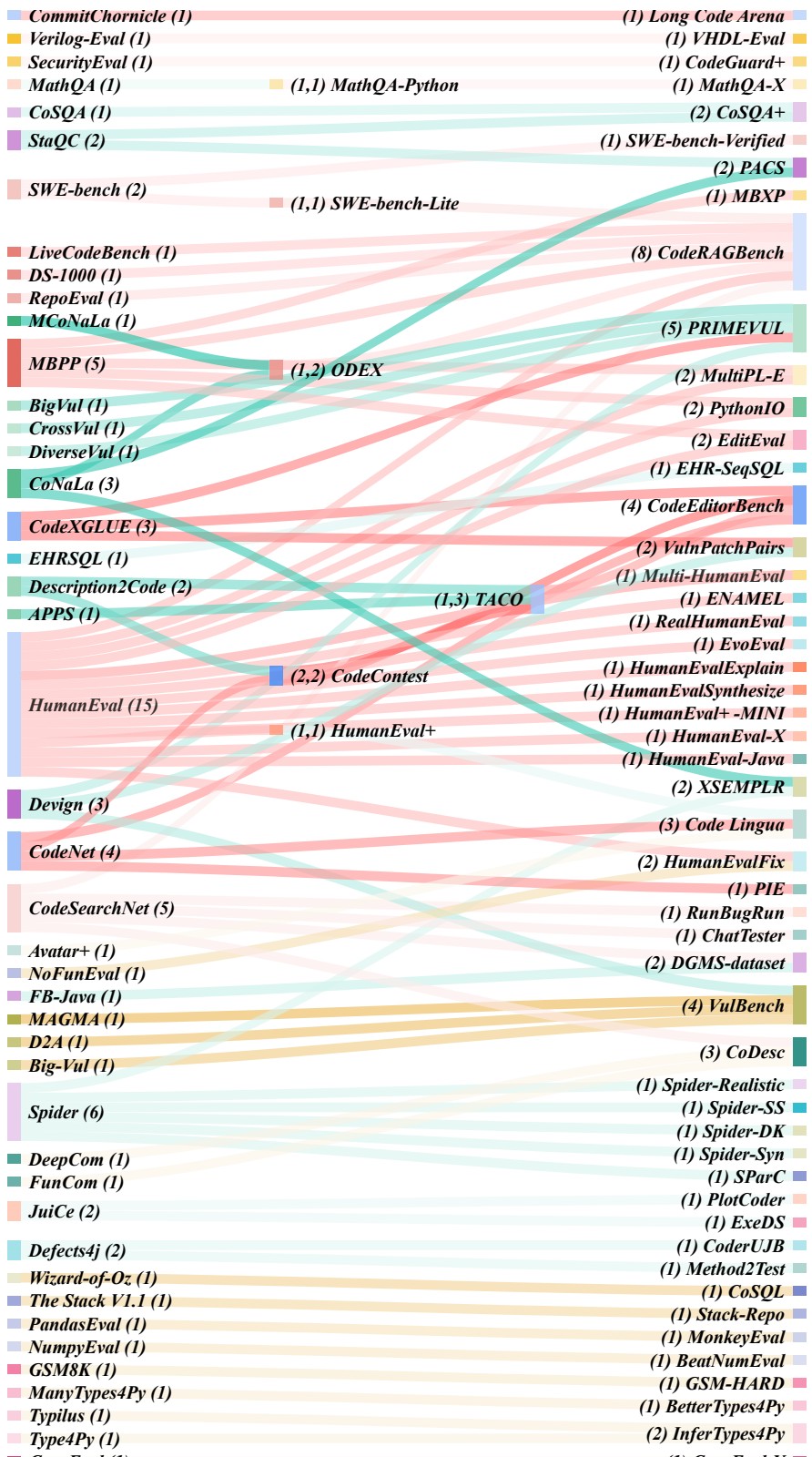

*Figure 59.* Relationships between Benchmarks

# D. Details of Human Study

## D.1. Interviewee Selection

The selection of interviewees is pivotal to ensuring the representativeness and relevance of the data collected. This involves identifying individuals with the knowledge or experience pertinent to the research theme.

To this end, we chose graduate students from SE or AI fields who have published at least one paper. This criterion ensures that participants have research experience and judgment capabilities. The focus on SE and AI fields is due to their likely interest in code benchmarks. Particularly, we aimed to recruit individuals who have published papers on code benchmarks to obtain firsthand feedback from experienced benchmark developers.

## D.2. Survey Question Design

**Questions.** The body of the survey was divided into five stages of benchmark development (following Figure 1), with necessary background information provided for each stage. Each criterion in HOW2BENCH was slightly modified to be in the first-person perspective, making it easier for interviewees to empathize and answer the questions from their own viewpoint. Finally, to facilitate comprehension, questions and instructions were translated into both English and Chinese.

**Question Setting**. To minimize the effort required from respondents, we designed ***single-choice questions*** with four options:

❏ I found it **important**, and I **have done** it.

❏ I found it **important**, although I **haven't done** it.

❏ I found it **not important**, but I **have done** it.

❏ I found it **not important**, and I **wouldn't** do it.

This format is intended to orthogonally explore the correlation between ***awareness*** and ***behavior***.

## D.3. Interview Process

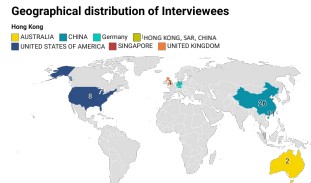

*Figure 60.* Geographical Distribution of Interviewees

**Questionnaire Distribution** The questionnaire was distributed via online platforms, targeting academic and professional networks related to SE and AI. *The distribution started on October 27, 2024, and ended on November 27th,*

*2024*, lasting one month.

**Results Collection** The responses were automatically collected through the online platform used for distribution.

**Survey Screening** Since the requirement was for participants who have published papers, responses from those selecting "No" to having published a paper were excluded. Also, incomplete surveys where not all questions were answered were also considered invalid and excluded from the analysis.

## D.4. Interview Result Analysis

In total, we collected 50 responses. The respondents were from seven regions, including the United States, the United Kingdom, Germany, Australia, China, and others, as shown in Figure 60. Only one survey was invalid due to the respondent selecting "have not published a paper", leaving **49 valid surveys** for analysis. A breakdown of the respondents' demographics is shown in Figure 61. The detailed responses for all 55 criteria in HOW2BENCH are shown in Figure 62 and Figure 63.



*Figure 61.* Demography of Interviewees

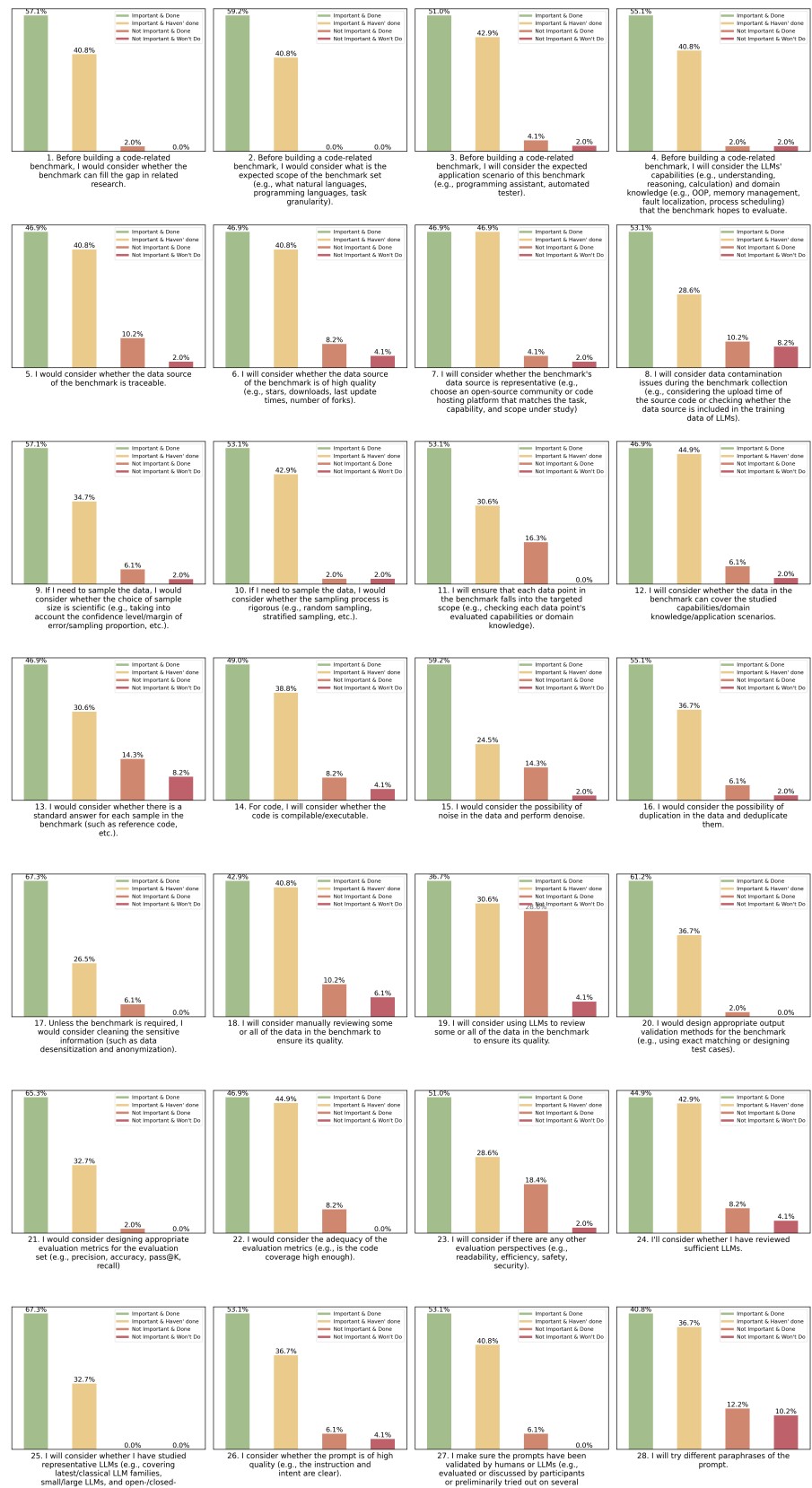

*Figure 62.* Results of Human Study (Questions 1 - 28

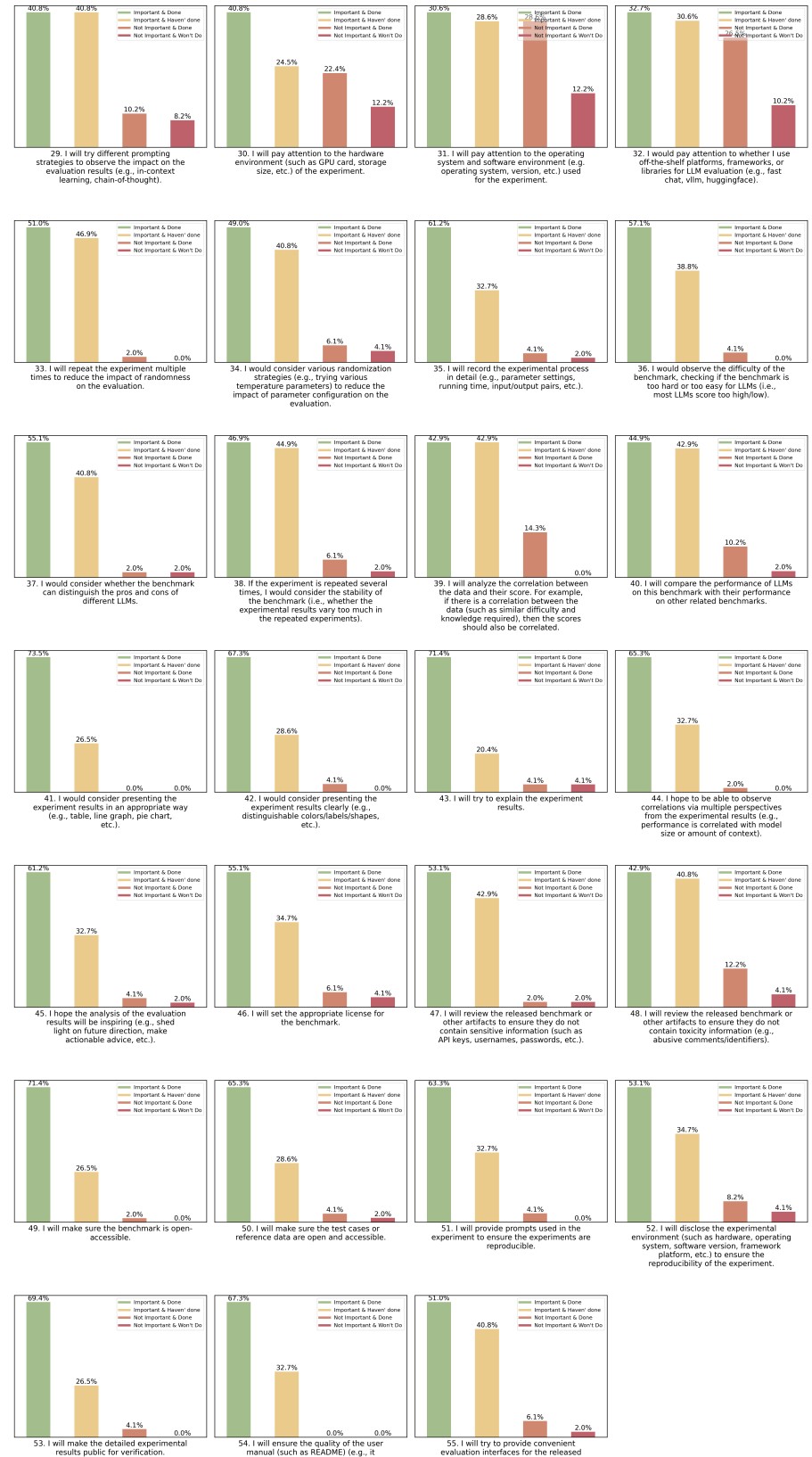

*Figure 63.* Results of Human Study (Questions 29 - 55)

# E. List of Studied Benchmarks (Focused Ones)

**Code Generation**: Five with top citations:

- HumanEval (Chen et al., 2021a)

- MBPP (Austin et al., 2021)

- CodeContest (Li et al., 2022)

- leetcodehardgym (Shinn et al., 2023)

- APPS (Hendrycks et al., 2021)

The latest one as of 31/12/2025:

- CIFE (Gunnu et al., 2025)

**Defect Detection**: Five with top citations:

- VulDeePecker (Li et al., 2018b)

- Devign (Zhou et al., 2019)

- Chromium and Debian (Chakraborty et al., 2022)

- $\mu$VulDeePecker (Zou et al., 2020)

- Synthetic Dataset (Hellendoorn et al., 2020)

The latest one as of 31/12/2025:

- Pushkar et al. (Pushkar et al., 2025a)

**Program Repair**: Five with top citations:

- Defects4J (Just et al., 2014)

- BFP (Tufano et al., 2019)

- MANYBUGS, INTROCLASS (Le Goues et al., 2015)

- HumanEval-Java (Jiang et al., 2023)

- QuixBugs (Prenner et al., 2022)

The latest one as of 31/12/2025:

- BackportBench (Zhong et al., 2025)

**Code Summarization**: Five with top citations:

- CODE-NN (Iyer et al., 2016)

- Java-small/med/large (Alon et al., 2019)

- code-summarization-public (Wan et al., 2018)

- HumanEvalPack (Muennighoff et al., 2024)

- Shrivastava et al. (Shrivastava et al., 2023b)

The latest one as of 31/12/2025:

- ContextCRBench (Hu et al., 2025a)

**Code Reasoning**: Five with top citations:

- LiveCodeBench (Jain et al., 2024)

- CRUXEval-X (Xu et al., 2024)

- xCodeEval (Khan et al., 2024)

- CodeQA (Liu & Wan, 2021)

- RepoQA (Liu et al., 2024b)

The latest one as of 31/12/2025:

- RE2-Bench (Liu et al., 2025a)

## F. List of Studied Benchmarks (Full)

We collected and studied 672 code-related benchmarks. We then listed and grouped them by year.

**2025**:

- Web-Bench (Xu et al., 2025b)
- EFFIBENCH-X (Qing et al., 2025)
- EDIT-Bench (Chi et al., 2025)
- CIRCLE (Chua, 2025)
- CS101-Gold (Havare et al., 2025)
- OSS-Bench (Jiang et al., 2025b)
- CodeArena (Du et al., 2025b)
- RefactorCoderQA (Rahman et al., 2025b)
- SwiftEval (Petrukha et al., 2025)
- CoRe (Xie et al., 2026)
- CodeElo (Quan et al., 2025)
- LongCodeBench (Rando et al., 2025)
- DSCodeBench (Ouyang et al., 2025)
- RustEvo$\hat{2}$ (Liang et al., 2025)
- JITVUL (Yildiz et al., 2025)
- UA-Code-Bench (Syromiatnikov & Ruvinskaya, 2025)
- CodeJudgeBench (Jiang et al., 2025a)
- HumanEval-Judge, MBPP-Judge, BigCodeBench-Judge (Yang et al., 2025a)
- ContextCRBench (Hu et al., 2025a)
- LoCoBench (Qiu et al., 2025)
- CodeMixBench (Sheokand & Sawant, 2025)
- VERINA (Ye et al., 2025)
- SecureAgentBench (Chen et al., 2025a)
- ClassEval-T (Xue et al., 2025)
- ProjectEval (Liu et al., 2025d)
- IFEvalCode (Yang et al., 2025d)
- CodeARC (Wei et al., 2025a)
- TestGenEval (Jain et al., 2025)
- CodePromptEval (Khojah et al., 2025)

- FreshBrew (May et al., 2025)
- CLOVER (Xu et al., 2025a)
- THROWBENCH (Prenner & Robbes, 2025)
- QuanBench (Guo et al., 2025b)
- FrontendBench (Zhu et al., 2025a)
- SolEval (Peng et al., 2025d)
- SWE-Bench-Live (Zhang et al., 2025c)
- SWE-Bench++ (Wang et al., 2025f)
- Defects4Log (Wang et al., 2025i)
- SecVulEval (Ahmed et al., 2025)
- PATCHEVAL (Wei et al., 2025b)
- VulnRepairEval (Wang et al., 2025h)
- FullStackBench (Bytedance-Seed-Foundation-Code-Team et al., 2025)
- CodeIF (Yan et al., 2025b)
- AutoCodeBench (Chou et al., 2025)
- TransLibEval (Xue et al., 2026)
- SWE-Perf (He et al., 2025a)
- TRACY (Gong et al., 2025)
- UnLeakedTestbench (Huang et al., 2025)
- TypyBench (Dong et al., 2025)
- LogicCat (Liu et al., 2025f)
- BIRD-INTERACT (Huo et al., 2025)
- SQL-Synth (Wang et al., 2025b)
- Falcon (Luo et al., 2025a)
- CORGI (Li et al., 2026)
- GITS-Eval (Rondon et al., 2025)
- CWEval (Peng et al., 2025a)
- DI-BENCH (Zhang et al., 2025e)
- LINUXFLBENCH (Zhou et al., 2025)
- ClassEval (Du et al., 2023)
- Deep-Bench (Daghighfarsoodeh et al., 2025)
- MCMD (Tao et al., 2022)

- HumanEvalNext (Koohestani et al., 2025b)
- FEA-Bench (Li et al., 2025c)
- SolBench (Chen et al., 2025d)
- DEFECTS4J-TRANS (Li et al., 2025a)
- CASTLE (Dubniczky et al., 2025)
- CodeIF-Bench (Wang et al., 2025g)
- Multi-SWE-bench (Zan et al., 2025)
- SciReplicate-Bench (Xiang et al., 2025)
- SWE-PolyBench (Rashid et al., 2025)
- APIRAT (Wang et al., 2025a)
- LeetCodeDataset (Xia et al., 2025)
- CoCo-Bench (Yin et al., 2025)
- SecRepoBench (Shen et al., 2025)
- CodeFlowBench (Wang et al., 2026)
- SWE-smith (Yang et al., 2025c)
- WebGen-Bench (Lu et al., 2025)
- SWE-rebench (Badertdinov et al., 2025)
- ResearchCodeBench (Hua et al., 2025)
- WebUIBench (Lin et al., 2025b)
- SWE-Factory (Guo et al., 2026)
- SafeGenBench (Li et al., 2025d)
- Zeng et al. (Zeng et al., 2025a)
- SWE-MERA (Pavel et al., 2025)
- F2STRANS (Zhang et al., 2025d)
- VulCoCo (Bui et al., 2025)
- NoCode-bench (Deng et al., 2025b)
- MRG-Bench (Li, 2025)
- STEPWISE-CODEX-Bench (Yan et al., 2025a)
- CodeFuse-CR-Bench (Guo et al., 2025a)
- SWE-Mirror (Wang et al., 2025e)
- SWE-Bench-Pro (Deng et al., 2025c)
- MultiSpider-2.0 (Pham et al., 2025)
- MULocBench (Zhang et al., 2025i)

- TC-Bench (Luo et al., 2025b)
- Defects4C (Wang et al., 2025d)
- GDPR-Bench-Android (Ran et al., 2025)
- PRDBench (Fu et al., 2025b)
- CodeAlignBench (Mehralian et al., 2025)
- Go-UT-Bench (Pipalani et al., 2025)
- RGym (Shehada et al., 2025)
- CodeFuse-CommitEval (Zhang et al., 2025g)
- PACIFIC (Dreyfuss et al., 2025)
- NL2Repo-Bench (Ding et al., 2026)
- RE2-Bench (Liu et al., 2025a)
- CIFE (Gunnu et al., 2025)
- SWE-EVO (Thai et al., 2026)
- Multi-Docker-Eval (Fu et al., 2025a)
- CodeCriticBench (Zhang et al., 2025a)
- Cui et al. (Cui et al., 2025)
- CodeReviewQA (Lin et al., 2025a)
- BigO(Bench) (Chambon et al., 2025)
- SWR-Bench (Zeng et al., 2025b)
- UniCode (Zheng et al., 2025b)
- SusVibes (Zhao et al., 2025a)
- COFFE (Peng et al., 2025c)
- DependEval (Du et al., 2025a)
- SWA-Bench, SWEE-Bench (Vergopoulos et al., 2025)
- CVE-Bench (Zhu et al., 2025b)
- Obscura (Nikiema et al., 2025)
- BinaryLLMs-Eval (Shang et al., 2025)
- CodeAssistBench (Kim et al., 2025)
- SWE-QA (Peng et al., 2025b)
- RECODE-H (Miao et al., 2025)
- CoReQA (Chen et al., 2025b)
- HumanEval-V (Wang et al., 2025c)
- CAMA (He et al., 2025b)

- C2RUST-BENCH (Sirlanci et al., 2025)

- CRUST-Bench (Khatry et al., 2025)

- OSVBench (Li et al., 2025b)

- VADER (Liu et al., 2025b)

- CodeSense (Roy et al., 2025)

- CETBench (Oza et al., 2025)

- DesignBench (Xiao et al., 2025)

- CoQuIR (Geng et al., 2025)

- MultiCodeIF (Duan et al., 2025)

- CoreCodeBench (Fu et al., 2026)

- WebMMU (Awal et al., 2025)

- ProjectAnalyzer (Gnieciak & Szandala, 2025)

- VulGate (Safdar et al., 2025)

- text2SQL4PM (Yamate et al., 2025)

- E2EDev (Liu et al., 2025c)

- Chart2Code (Tang et al., 2026)

- RealClassEval (Rahman et al., 2025a)

- BackPortBench (Zhong et al., 2025)

- Sun et al. (Sun et al., 2025a)

- Pushkar et al. (Pushkar et al., 2025b)

- RepoTransBench (Wang et al., 2025j)

- DomainCodeBench (Zheng et al., 2025a)

- UniVul (Sun et al., 2025b)

- mHumanEval (Raihan et al., 2025)

- WebCode2M (Gui et al., 2025)

- CodeJudge-Eval (Zhao et al., 2025b)

- HumanEval-XL (Peng et al., 2024)

- Visual-SWE-bench (Zhang et al., 2025f)

**2024**:

- CodeEditorBench (Guo et al., 2024)

- MHPP (Dai et al., 2024)

- LiveCodeBench (Jain et al., 2024)

- CodeAgentBench (Zhang et al., 2024b)

- CruxEval (Gu et al., 2024)

- BigCodeBench (Zhuo et al., 2024)

- OOPEval (Wang et al., 2024c)

- DevEval (Li et al., 2024a)

- Long Code Arena (Bogomolov et al., 2024)

- CodeRAGBench (Wang et al., 2024f)

- ScenEval (Paul et al., 2024)

- AICoderEval (Xia et al., 2024b)

- VersiCode (Wu et al., 2024b)

- VHDL-Eval (Vijayaraghavan et al., 2024)

- NaturalCodeBench (Zhang et al., 2024d)

- CodeGuard+ (Fu et al., 2024)

- PECC (Haller et al., 2024)

- USACO (Shi et al., 2024b)

- ParEval (Nichols et al., 2024)

- MxEval (Athiwaratkun et al., 2022)

- MMCode (Li et al., 2024c)

- Plot2Code (Wu et al., 2024a)

- ChartMimic (Shi et al., 2024a)

- DebugBench (Tian et al., 2024)

- PythonIO (Zhang et al., 2024e)

- StaCCQA (Yang et al., 2024a)

- RepoQA (Liu et al., 2024b)

- PRIMEVUL (Ding et al., 2024a)

- VulDetectBench (Liu et al., 2024d)

- ProCQA (Li et al., 2024e)

- CoSQA+ (Gong et al., 2024a) (Huang et al., 2021)

- JavaBench (Cao et al., 2024a)

- HumanEvo (Zheng et al., 2024)

- REPOEXEC (Hai et al., 2024)

- EHR-SeqSQL (Ryu et al., 2024)

- BookSQL (Kumar et al., 2024)

- AMBROSIA (Saparina & Lapata, 2024)

- WUB, WCGB (Yun et al., 2024)
- RES-Q (LaBash et al., 2024)
- PythonSaga (Yadav et al., 2024a)
- Mercury (Du et al., 2024)
- ENAMEL (Qiu et al., 2024a)
- RealHumanEval (Mozannar et al., 2024)
- CoderUJB (Zeng et al., 2024)
- EvoEval (Xia et al., 2024a)
- ML-Bench (Liu et al., 2023c)
- VerilogEval (Pinckney et al., 2024)
- CodeApex (Fu et al., 2023)
- HumanEvalPack (Muennighoff et al., 2024)
- HumanEval+ (Liu et al., 2023a)
- HumanEval-X (Zheng et al., 2023a)
- XCodeEval (Khan et al., 2024)
- CoderEval (Yu et al., 2023)
- CodeXGLUE (Lu et al., 2021)
- VulnPatchPairs (Risse & Böhme, 2024)
- WikiSQL (Zhong et al., 2017)
- CrossCodeEval (Ding et al., 2023)
- SWE-bench (Jimenez et al., 2024)
- BAIRI et al. (Bairi et al., 2024)
- BioCoder (Tang et al., 2024)
- RepoBench (Liu et al., 2024c)
- NoFunEval (Singhal et al., 2024)
- CoCoMIC (Ding et al., 2024b)
- Java-small/med/large (Alon et al., 2019)
- FixEval (Haque et al., 2023)
- CommitBench (Schall et al., 2024)
- InfiAgent-DABench (Hu et al., 2024)
- InfiBench (Li et al., 2024d)
- Design2Code (Si et al., 2024)
- MatPlotBench (Yang et al., 2024c)

- EditEval (Li et al., 2024b)
- D1, D2, D3 (Huang et al., 2024)
- RepoEval (Liao et al., 2024)
- BetterTypes4Py, InferTypes4Py (Wei et al., 2023)
- HumanEval-Java (Jiang et al., 2023)
- PIE (Shypula et al., 2024)
- EvalGPTFix (Zhang et al., 2023a)
- EHRSQL (Lee et al., 2023)
- Spider2-V (Cao et al., 2024c)
- TESTEVAL (Wang et al., 2024d)
- ChatTester (Yuan et al., 2023a)
- Code Lingua (Pan et al., 2024a)
- EffiBench (HUANG et al., 2024)
- CRUXEval-X (Xu et al., 2024)
- DomainEval (Zhu et al., 2024)
- SWE-Bench+ (Aleithan et al., 2024)
- eyeballvul (Chauvin, 2024)
- ComplexCodeEval (Feng et al., 2024)
- BabelBench (Wang et al., 2024e)
- CRQBench (Dinella et al., 2024)
- R2C2-Coder (Deng et al., 2025a)
- SAFIM (Gong et al., 2024b)
- McEval (Chai et al., 2024)
- PolyHumanEval (Tao et al., 2024)
- RustRepoTrans (Ou et al., 2025)
- RAPID (Ma et al., 2024)
- BEAVER (Chen et al., 2025c)
- Spider2.0 (Lei et al., 2025)
- Wu et al. (Wu et al., 2024c)
- BigTable-0.2k (Zhang et al., 2024a)
- SWT-Bench (Mündler et al., 2025)
- CodeSecEval (Wang et al., 2024b)
- SWE-bench-java (Zan et al., 2024)

- TypeEvalPy/PyCG (Venkatesh et al., 2025)

- M²RC-EVAL (Liu et al., 2024a)

- MdEval (Liu et al., 2025e)

- CPP-UT-Bench (Bhargava et al., 2024)

- OBFUSEVAL (Zhang et al., 2025h)

- ExecRepoBench (Yang et al., 2024b)

- VulnLLMEval (Zibaeirad & Vieira, 2024)

- TestBench (Zhang et al., 2024c)

- Codev-Bench (Pan et al., 2024b)

- SWE-bench Multimodal (Yang et al., 2025b)

- HumanEval-V (Zhang et al., 2025b)

- HumanEval_T (Bradbury & More, 2025)

- TDD-Bench Verified (Ahmed et al., 2024)

**2023**:

- MCoNaLa (Wang et al., 2023b)

- MultiPL-E (Cassano et al., 2022)

- ODEX (Wang et al., 2022)

- TACO (Li et al., 2023b)

- DOTPROMPTS, MGDMICROBENCH (Agrawal et al., 2023)

- StudentEval (Babe et al., 2024)

- CodeTransOcean (Yan et al., 2023)

- G-TransEval (Jiao et al., 2023)

- AVATAR (Ahmad et al., 2023)

- RunBugRun (Prenner & Robbes, 2023)

- VulBench (Gao et al., 2023b)

- DiverseVul (Chen et al., 2023)

- Hellendoorn et al. (Hellendoorn et al., 2020)

- XSemPLR (Zhang et al., 2023b)

- BIRD (Li et al., 2023a)

- Stack-Repo (Shrivastava et al., 2023a)

- RepoEval (Liao et al., 2024)

- MTPB (Nijkamp et al., 2022)

- ARCADE (Yin et al., 2023)

- Shrivastava et al. (Shrivastava et al., 2023b)

- Grag et al. (Garg et al., 2022)

- GSM-HARD (Gao et al., 2023a)

- InferredBugs (Jin et al., 2023)

- LeetcodeHardGym (Shinn et al., 2023)

- APIBench (Patil et al., 2023)

- ClassEval (Du et al., 2023)

- CommitChronicle (Eliseeva et al., 2023)

- TeCo (Nie et al., 2023)

- TESTPILOT (Schäfer et al., 2024)

**2022**:

- AixBench (Hao et al., 2022)

- TypeBugs (Oh & Oh, 2022)

- XLCoST (Zhu et al., 2022)

- CS1QA (Lee et al., 2022)

- Chromium and Debian (Chakraborty et al., 2022)

- Spider-Realistic (Deng et al., 2021)

- Spider-SS (Gan et al., 2022)

- DSP (Chandel et al., 2022)

- CodeContest (Li et al., 2022)

- PandasEval, NumpyEval (Zan et al., 2022b)

- TorchDataEval, MonkeyEval, BeatNumEval (Zan et al., 2022a)

- DS-1000 (Lai et al., 2023)

- MCMD (Tao et al., 2022)

- ExeDS (Huang et al., 2022)

- QuixBugs (Prenner et al., 2022)

- ManyTypes4Py v0.7 (Mir et al., 2022)

- Lyra (Liang et al., 2022)

**2021**:

- SySeVR (Li et al., 2018a)

- Ling&Wu et al. (Ling et al., 2021)

- Chen et al. (Chen et al., 2021b)

- MBPP, MathQA-Python (Austin et al., 2021)

- HumanEval (Chen et al., 2021a)

- APPS (Hendrycks et al., 2021)

- Berabi et al. (Berabi et al., 2021)

- CrossVul (Nikitopoulos et al., 2021)

- PYPIBUGS, RANDOMBUGS (Allamanis et al., 2021)

- D2A (Zheng et al., 2021)

- CodeQA (Liu & Wan, 2021)

- Spider-DK (Gan et al., 2021b)

- KaggleDBQA (Lee et al., 2021)

- SEDE (Hazoom et al., 2021)

- Spider-Syn (Gan et al., 2021a)

- CoDesc (Hasan et al., 2021)

- Methods2Test (Tufano et al., 2022)

- Rozière et al. (Rozière et al., 2022)

**2020**:

- Lachaux&Roziere et al. (Rozière et al., 2020)

- $\mu$VulDeePecker (Zou et al., 2020)

- CosBench (Yan et al., 2020)

- PACS (Heyman & Cutsem, 2020)

- Criteria2SQL (Yu et al., 2020)

- SQUALL (Shi et al., 2020)

- Hu et al. (Hu et al., 2019)

- CodeSearchNet Challenge (Husain et al., 2019)

- MIMICSQL (Wang et al., 2020)

- Atlas (Watson et al., 2020)

- Liu et al. (Liu et al., 2022)

- Android (Agarwal et al., 2020)

- CCSD (Liu et al., 2021)

**2019**:

- BFP (Tufano et al., 2019)

- SARD (Lin et al., 2019)

- Spider (Yu et al., 2018)

- JuICe (Agashe et al., 2019)

- Nguyen et al. (Nguyen et al., 2019)

- Lin et al. (Lin et al., 2021)

- Zhou et al. (Zhou et al., 2019)

- CoSQL (Yu et al., 2019a)

- SParC (Yu et al., 2019b)

- Malik (Malik et al., 2019)

- LeClair (LeClair et al., 2019)

**2018**:

- CoNaLa (Yin et al., 2018)

- DeepCom (Hu et al., 2018a)

- TL-CodeSum (Hu et al., 2018b)

- code-summarization-public (Wan et al., 2018)

- Russell et al. (Russell et al., 2018)

- VulDeePecker (Li et al., 2018b)

- Lin et al. (Lin et al., 2018)

- StaQC (Yao et al., 2018)

- Advising (Finegan-Dollak et al., 2018)

- ConCode (Iyer et al., 2018)

- NNGen (Liu et al., 2018)

- Gu et al. (Gu et al., 2018)

**2017**:

- QuixBugs (Lin et al., 2017)

- the DeepFix dataset (Gupta et al., 2017)

- Barone et al. (Barone & Sennrich, 2017)

**2016**:

- CODE-NN (Iyer et al., 2016)

- Mou et al. (Mou et al., 2016)

**2015**:

- MANYBUGS, INTROCLASS (Le Goues et al., 2015)

**2014**:

- Defects4j (Just et al., 2014)

- BigCloneBench (Svajlenko et al., 2014)

## G. Guideline

Finally, for ease of printing and use, we organized the guideline HOW2BENCH into a clear, color-coded checklist (4 pages in total) that is easy to print, attached at the end of the paper.

| | | Priority | ☑ |
|---|---|---|---|
| | **Phase 1. Benchmark Design** | | |
| 1 | **Consider whether the benchmark can fill the gap in related research.** | ★★★ | ☐ |
| 2 | **Consider what is the expected scope of the benchmark set** (e.g., what natural languages, programming languages, task granularity)**.** | ★★★ | ☐ |
| 3 | **Consider the expected application scenario of this benchmark** (e.g., programming assistant, automated tester). | ★★★ | ☐ |
| 4 | **Consider the LLMs' capabilities** (e.g., understanding, reasoning, calculation) **and domain knowledge** (e.g., OOP, memory management, fault localization, process scheduling) **that the benchmark hopes to evaluate.** | ★★★ | ☐ |
| | **Phase 2. Benchmark Construction (1/2)** | Priority | ☑ |
| 5 | **Consider whether the data source of the benchmark is traceable.** | ★ | ☐ |
| 6 | **Consider whether the data source of the benchmark is of high quality** (e.g., stars, downloads, last update times, number of forks). | ★★ | ☐ |
| 7 | **Consider whether the benchmark's data source is representative** (e.g., choose an open-source community or code hosting platform that matches the task, capability, and scope under study) | ★ | ☐ |
| 8 | **Consider data contamination issues during the benchmark collection** (e.g., considering the upload time of the source code or checking whether the data source is included in the training data of LLMs). | ★ | ☐ |
| 9 | **If data sampling is needed, consider whether the choice of sample size is scientific** (e.g., considering the confidence level/margin of error/sampling proportion, etc.). | ★ | ☐ |
| 10 | **If data sampling is needed, consider whether the sampling process is rigorous** (e.g., random sampling, stratified sampling, etc.). | ★ | ☐ |
| 11 | **Ensure each data point in the benchmark falls into the targeted scope** (e.g., checking each data point's evaluated capabilities or domain knowledge). | ★ | ☐ |
| 12 | **Consider whether the data in the benchmark can cover the studied capabilities/domain knowledge/application scenarios.** | ★★★ | ☐ |
| 13 | **Consider whether there is a standard answer for each sample in the benchmark** (such as reference code, etc.). | ★★ | ☐ |
| 14 | **For code, consider whether the code is compilable/executable.** | ★ | ☐ |
| 15 | **Consider the possibility of noise in the data and perform denoise.** | ★★ | ☐ |
| 16 | **Consider the possibility of duplication in the data and deduplicate them.** | ★★ | ☐ |
| 17 | **Clean the sensitive information** (such as data desensitization and anonymization) **unless the benchmark is deliberately designed so.** | ★★ | ☐ |

| | Phase 2. Benchmark Construction (2 / 2) | Priority | ☑ |
|---|---|---|---|
| 18 | Manually review some or all of the data in the benchmark to ensure its quality. | ★★ | ☐ |
| 19 | Use LLMs to review some or all of the data in the benchmark to ensure its quality. | ★ | ☐ |
| 20 | Design appropriate output validation methods for the benchmark (e.g., using exact matching or designing test cases). | ★★★ | ☐ |
| 21 | Design appropriate evaluation metrics for the evaluation set (e.g., precision, accuracy, pass@K, recall). | ★★★ | ☐ |
| 22 | Consider the adequacy of the evaluation metrics (e.g., is the code coverage high enough). | ★★★ | ☐ |
| 23 | Consider if there are any other evaluation perspectives (e.g., readability, efficiency, safety, security). | ★ | ☐ |
| | Phase 3. Benchmark Evaluation | Priority | ☑ |
| 24 | Consider whether sufficient LLMs are evaluated. | ★ | ☐ |
| 25 | Consider whether representative LLMs (e.g., covering latest/classical LLM families, small/large LLMs, and open-/closed-source LLMs) **are evaluated.** | ★ | ☐ |
| 26 | Consider whether the prompt is of high quality (e.g., the instruction and intent are clear). | ★★★ | ☐ |
| 27 | The prompts have been validated by humans or LLMs (e.g., evaluated or discussed by participants or preliminarily tried out on several LLMs). | ★ | ☐ |
| 28 | Try different paraphrases of the prompt. | ★ | ☐ |
| 29 | Try different prompting strategies to observe the impact on the evaluation results (e.g., in-context learning, chain-of-thought). | ★★ | ☐ |
| 30 | Pay attention to the hardware environment (such as GPU card, storage size, etc.) of the experiment. | ★★★ | ☐ |
| 31 | Pay attention to the operating system and software environment (e.g. operating system, version, etc.) **used for the experiment.** | ★★★ | ☐ |
| 32 | Pay attention to the off-the-shelf platforms, frameworks, or libraries for LLM evaluation (e.g., fast chat, vllm, huggingface) **that are used.** | ★★ | ☐ |
| 33 | Repeat the experiment multiple times to reduce the impact of randomness on the evaluation. | ★ | ☐ |
| 34 | Consider various randomization strategies (e.g., trying various temperature parameters) **to reduce the impact of parameter configuration on the evaluation.** | ★★ | ☐ |
| 35 | Record the experimental process in detail (e.g., parameter settings, running time, input/output pairs, etc.). | ★★★ | ☐ |

| | Phase 4. Benchmark Analysis | Priority | ☑ |
|---|---|---|---|
| 36 | **Observe the difficulty of the benchmark, checking if the benchmark is too hard or too easy for LLMs** (i.e., most LLMs score too high/low). | ★★ | ☐ |
| 37 | **Consider whether the benchmark can distinguish the pros and cons of different LLMs.** | ★ | ☐ |
| 38 | **If the experiment is repeated several times, consider the stability of the benchmark** (i.e., whether the experimental results vary too much in the repeated experiments). | ★ | ☐ |
| 39 | **Analyze the correlation between the data and their score**. For example, if there is a correlation between the data (such as similar difficulty and knowledge required), then the scores should also be correlated. | ★★ | ☐ |
| 40 | Compare the performance of LLMs on this benchmark with their performance on **other related benchmarks**. | ★ | ☐ |
| 41 | **Consider presenting the experiment results in an appropriate way** (e.g., table, line graph, pie chart, etc.). | ★★★ | ☐ |
| 42 | **Consider presenting the experiment results clearly** (e.g., distinguishable colors/labels/shapes, etc.). | ★★★ | ☐ |
| 43 | **Explain the experiment results.** | ★★★ | ☐ |
| 44 | **Observe correlations via multiple perspectives from the experimental results** (e.g., performance is correlated with model size or amount of context). | ★ | ☐ |

| | **Phase 5. Benchmark Release** | **Priority** | ☑ |
|---|---|---|---|
| 45 | **The analysis of the evaluation results will be inspiring** (e.g., shed light on future direction, make actionable advice, etc.). | ★ | ☐ |
| 46 | **Set the appropriate license for the benchmark.** | ★★★ | ☐ |
| 47 | **Review the released benchmark or other artifacts to ensure they do NOT contain sensitive information** (e.g., API keys, usernames, passwords, etc.). | ★★★ | ☐ |
| 48 | **review the released benchmark or other artifacts to ensure they do NOT contain toxicity information** (e.g., abusive comments/identifiers). | ★★★ | ☐ |
| 49 | **Make sure the benchmark is open-accessible.** | ★★★ | ☐ |
| 50 | **Make sure the test cases or reference data are open and accessible.** | ★★★ | ☐ |
| 51 | **Provide prompts used in the experiment to ensure the experiments are reproducible.** | ★★★ | ☐ |
| 52 | **Disclose the experimental environment** (e.g., hardware, operating system, software version, framework platform) **to ensure the reproducibility of the experiment.** | ★★★ | ☐ |
| 53 | **Make the detailed experimental results public for verification.** | ★★★ | ☐ |
| 54 | **Ensure the quality of the user manual** such as README (e.g., it contains necessary benchmark introduction, executable scripts, etc.). | ★★ | ☐ |
| 55 | **Provide convenient evaluation interfaces for the released benchmark** (e.g., providing a command line interface, docker, etc.). | ★★ | ☐ |