# OpenReview forum: "Position: Code Benchmarks Should Prioritize Rigor, Reliability, and Reproducibility"
_ICML.cc/2026/Position_Paper_Track — ICML 2026 Position Paper Track regular_

### Official Review · Reviewer_qktb · 2026-02-18

**Significance:** 2
**Argument Clarity:** 3
**Rating:** 4
**Confidence:** 4

**Questions:**

1. What is the detailed steps to obtain the statistics?

2. Can you provide some in-depth analysis on the new popular benchmarks?

**Alternative Views Section:**

Yes

**Compliance With Llm Reviewing Policy A Conservative:**

Affirmed.

**Discussion Potential:**

2

**Final Justification:**

The authors response addressed my concerns, I keep my positive score on this paper.

**Paper Summary:**

### Summary
This paper argues that code-related benchmarks for evaluating large language models (LLMs) should emphasize rigor in construction, reliability in evaluation, and reproducibility in release. The authors conduct a large-scale survey of 572 code benchmarks published between 2014 and 2025 and introduce a structured evaluation framework. They operationalize their proposal through HOW2BENCH, a 55-item checklist designed to guide benchmark developers and reviewers. The paper also includes (1) a focused case study of 30 representative benchmarks and (2) a questionnaire-based human study with 49 researchers to examine awareness gaps and perceived trade-offs between rigor and efficiency.

**Position:**

Yes

**Position In Title:**

Yes

**Related Work:**

2

**Strengths And Weaknesses:**

## Strengths

1. Timely and Important Topic. The paper addresses a highly relevant issue. With the rapid proposal of code benchmarks, concerns about contamination, evaluation reliability, and reproducibility are indeed pressing.

2. Large-Scale Empirical Profiling. This paper Surveyes 572 benchmarks over a decade is ambitious and valuable. The longitudinal statistics (e.g., code coverage omission, data contamination neglect, lack of reproducibility artifacts) provide concrete evidence supporting the authors’ claims. Although it is unclear how these statisticales are obtained.


## Weaknesses


### 1. Position Paper vs. Survey

This paper reads more like: A large-scale survey of benchmark practices, or A checklist-style guideline proposal instead of a position paper aruging a new arugement.
As the core idea—benchmarks should be rigorous, reliable, and reproducible—is not new. Similar concerns have been raised repeatedly in exitsing work.
The main contribution appears to be the consolidation of these concerns into a 55-item checklist combined with descriptive statistics.
However, the design principles behind the checklist are not clearly justified. It is unclear what criteria were used to determine the inclusion, granularity, and independence of the 55 items. The conceptual contribution therefore feels incremental. (For example, I check the creteria 49 and 50, 49 states benchmark should be public accessiable, and 50 states test cases should be public accessiable, which seems to be redundant, as test cases are part of the benchmark)


### 2. Studied Benchmarks Are Already Outdated

According to Appendix E (as stated in the paper), the selected datasets and benchmarks are limited to those published before 31/08/2024. Given that the submission targets ICML 2026, the dataset selection cutoff is already outdated. Benchmark construction practices evolve extremely rapidly in the LLM era. Excluding late-2024 and 2025 benchmarks from the representative case study weakens the claims about current practice.

Moreover, the paper largely discusses benchmarks in terms of code generation benchmarks, more complex code-related benchmarks are under-discussed or not clearly analyzed, such as SWE-Bench are not discussed in details， just cited in the paper. I


### 3. Insufficient Methodological Transparency for Statistics

One of the most critical issues concerns the statistical profiling methodology.

The paper reports numerous statistics, However, the following details remain unclear: what is the detailed steps to get these numbers?
For example, this paper claim only 31.6% benchmarks have deduplicated the collected data. How is this number obtained, did the auhtors try to examine the dataset and examine whether there are deduplicated data in the dataset?
Same issue for other statistic numbers. Given that the central claims rely heavily on these statistics, the lack of detailed methodology undermines credibility. A paper advocating rigor must itself demonstrate rigorous measurement methodology.



### 4. “Alternative View” Is Not a Standalone Section

The “Alternative View” discussion is embedded within earlier sections rather than structured as a clearly defined standalone section.


### 5. Section repeatation. Appendix B. Detailed Guidance in HOW2BENCH, seems to appear twice in the paper.


## Missing a lot related work, more related work could be found in  `https://github.com/tongye98/Awesome-Code-Benchmark`

GitTaskBench: A Benchmark for Code Agents Solving Real-World Tasks Through Code Repository Leveraging

DyCodeEval: Dynamic Benchmarking of Reasoning Capabilities in Code Large Language Models Under Data Contamination


TRACY: Benchmarking Execution Efficiency of LLM-Based Code Translation

BinMetric: A Comprehensive Binary Analysis Benchmark for Large Language Models

Dynamic Benchmark Construction for Evaluating Large Language Models on Real-World Codes

AppForge: From Assistant to Independent Developer -- Are GPTs Ready for Software Development?

A.S.E: A Repository-Level Benchmark for Evaluating Security in AI-Generated Code

PPM: Automated Generation of Diverse Programming Problems for Benchmarking Code Generation Models


Exploring Language Model's Code Generation Ability with Auxiliary Functions

DynaCode: A Dynamic Complexity-Aware Code Benchmark for Evaluating Large Language Models in Code Generation

CPRet: A Dataset, Benchmark, and Model for Retrieval in Competitive Programming


ELABORATION: A Comprehensive Benchmark on Human-LLM Competitive Programming

OIBench: Benchmarking Strong Reasoning Models with Olympiad in Informatics

HumanEval Pro and MBPP Pro: Evaluating Large Language Models on Self-invoking Code Generation

**Support:**

3

---

> ### Author Rebuttal · Authors · 2026-03-31
>
> ## Q1. Detailed-steps-to-obtain-the-statistics?
>
> **Annotation-process**
>
> To further clarify the annotation workflow beyond the high‑level description above, we summarize the concrete steps we followed when labeling the 572 benchmarks:
>
> 1. **Independent annotation by two primary annotators**
>
>     For each benchmark paper, two primary annotators independently:
>     1. Skimmed the full paper and then carefully read sections likely to contain relevant information (e.g., Dataset/Data, Data Collection, Data Curation/Cleaning, Quality Control/Assurance).
>     2. Assigned each attribute a label from {present, not mentioned}.
>     3. Recorded textual evidence (exact sentences or locations) whenever an attribute was labeled as “present,” storing all evidence in a shared annotation sheet.
>
> 2. **Conflict detection and resolution with additional annotators**
>     After both primary annotators completed their labels, we automatically identified conflicts (e.g., one annotator labeled "has contamination check" as present while the other labeled it as absent).
>
>     For each conflicted benchmark–attribute pair:
>         - The two primary annotators first re‑checked the paper and their evidence.
>         - If disagreement remained, two additional annotators joined the discussion to reach a consensus
>
> 3. **Finalization and quality control**
>     - Only consensus labels were used in our descriptive statistics.
>     - We calculated the **inter-annotator agreement**: For the inter-annotator Cohen’s kappa between the two primary annotators ranged from 0.76 to 0.92 across different attributes.
>
> ### Q1(b). Criteria-and-keyword‑based-evidence-for-attribute-annotation-to-obtain-the-statistics
> To make annotations as objective and reproducible as possible, annotators relied on explicit textual cues:
>
> - **Criteria-for-Checking-whether-Data-quality-assurance-present**
>     > **Criteria**: We labeled a benchmark as "considers data quality assurance" only if we identified a dedicated section, subsection, or explicit sentences describing how data quality was ensured.
>
>     Typical evidence includes sections or sentences tagged as **"Data Curation / Cleaning / Preprocessing", "Quality Control / Quality Assurance"**, etc., or detailed descriptions of construction and filtering procedures.
>
> - **Criteria-for-Type-of-data-quality-assurance: Manual check**:
>
>     > **Criteria**: We mark a benchmark as using manual checks if it explicitly mentions human involvement with **keywords** such as:
>
>     - "manual", "human", "inspection", "verification", "review"
>     - "spot check", "random sampling", "audt", "sanity check"
>     - "hand‑curated", "human‑in‑the‑loop", "expert"
>     - "label correction", "error analysis", "case‑by‑case"
>
> - **Type-of-data-quality-assurance: Code execution check**
>     > **Criteria**: We mark a benchmark as using code execution checks if it describes validating items with **keywords** such as:
>     - "execute/execution", "run", "compile", "build"
>     - "unit tests", "test suite"
>     - "pass/fail"
>
> - **Type-of-data-quality-assurance-LLM‑as‑judge**:
>
>     > **Criteria**: We mark a benchmark as using LLM‑as‑judge with phrases like:
>
>     - "LLM‑as‑a‑judge", "LLM judge"
>     - "model judge"
>     - "[model name] as judge"
>
> - **Checking Data contamination check**
>
>     > **Criteria**: We mark a benchmark as checking for contamination if it explicitly examines or mitigates training–test overlap. We rely on terms such as:
>
>     - "data contamination", "leakage"
>     - "overlap"
>     - "Jaccard similarity"
>     - "time‑based split",
>     - "cutoff date", "post‑YYYY"
>
> - **Criteria for Checking Data deduplication**
>
>     > **Criteria**: We mark deduplication as present if the benchmark explicitly states with cues like:
>
>     - "deduplication", "de‑dup"
>     - "remove duplicates", "duplicate removal"
>     - "redundant", "repetitive"
>
> **Criteria-for-Checking Coverage considerations**
>
> For code coverage, we look for mentions of:
>
> - "coverage"
> - "test adequacy", "test strength", "test suite quality"
> - "corner case(s)" or related phrasing
>
> ## Q2. More analysis on new benchmarks
>
> Thanks for the suggestion. We managed to **add 100 more new/missing code benchmarks** during rebuttal.
>
> Key statistics before/after updates:
>
> ● 46.0% -> *45.9%* benchmarks did not go through quality assurance check; 82.5% -> **79.7%** did not consider data contamination; 62% -> **67.1%** did not deduplicate the data points;
>
> ● 84.2% -> **85.0%** did not ensure a reliable judgement, such as ensuring a high code coverage when test suites are provided; 64.0% -> **66.1%** benchmark evaluation was one-pass, without repeating the experiment to avoid randomness;
>
> ● 38.8% -> **38.2%** of the benchmarks did not provide the essential information (e.g., prompts) for reproducibility; 80.0% -> **80.0%** (same) did not provide the log information on the benchmarks; 12.4% -> **14.7%** are not open source, 2.6% -> **2.2%** only partially released;
>
> We will update all the statistics in the final version.

---

> > ### Author Rebuttal · Reviewer_qktb · 2026-04-03
> >
> > Thanks for the response. Based on the authors’ description, the criteria appear to rely primarily on keyword matching, which raises concerns about the method’s effectiveness. For example, in the case of data deduplication, it is not necessary to explicitly mention keywords such as SWE-Bench for a benchmark to still contain non-deduplicated data. This makes me question whether the reported numbers in the paper may be overestimated.
> >
> > I also examined Criteria 49 and 50. Criterion 49 states that the benchmark should be publicly accessible, while Criterion 50 requires that the test cases be publicly accessible. These seem somewhat redundant, since test cases are inherently part of a benchmark. How do the authors distinguish between these two criteria, and how do they ensure consistency in their evaluation?
> >
> > More importantly, I would like the authors to address my key concern: the paper reads more like a survey than a position paper.

---

### Official Review · Reviewer_cFaa · 2026-03-12

**Significance:** 4
**Argument Clarity:** 2
**Rating:** 4
**Confidence:** 4

**Questions:**

Why are the QA-check statistics inconsistent across the paper? Please report the denominator for each statistic, the exact definition of “QA check” (i.e., what procedures qualify), and which subset it is computed over (all 572 benchmarks vs. a subsample). Concretely, on page 6 the paper states both “48.1% did not go through QA” and “only 31% went through a QA check,” which are difficult to reconcile without clarifying scope and definitions.
	Quality control for annotation and statistics: Were multiple annotators involved? What inter-annotator agreement (IAA) was achieved? How were disagreements adjudicated and what was the protocol for resolving ambiguous cases?
	Selection-bias check for the “representative 30 benchmarks” subset: Several analyses rely on a representative subsample. Please specify the selection criteria and provide a representativeness test—e.g., whether the subset matches the full set of 572 in terms of year, task type, venue, and influence/visibility. Without such validation, conclusions drawn from the subset should not be generalized.
	Contamination detection methodology and definition: When reporting that “82.5% did not consider contamination,” what is the operational definition of contamination (e.g., train–test leakage, near-duplicate overlap, benchmark–pretraining overlap)? Could you run at least two complementary contamination checks on selected benchmarks (e.g., string/hash-based near-duplicate detection plus embedding-based semantic similarity detection) and compare the detected contamination rates to quantify what “not considering contamination” implies in terms of actual risk?

**Alternative Views Section:**

Yes

**Compliance With Llm Reviewing Policy A Conservative:**

Affirmed.

**Discussion Potential:**

4

**Final Justification:**

The rebuttal addressed my main concerns. I have no further questions and will raise my score.

**Paper Summary:**

This is a position paper arguing that code benchmarks should prioritize rigor, reliability, and reproducibility over rapid release cycles or leaderboard-driven incentives. The authors support this stance with a longitudinal survey of 572 benchmarks from 2014–2025, showing that key quality practices remain widely missing in current benchmarking workflows. They further propose HOW2BENCH, a checklist-style framework covering the full benchmark lifecycle, consisting of 55 prioritized criteria, and complement it with a small-scale human study (survey/interviews) suggesting that both resource constraints (cost, time, effort) and an awareness gap contribute to the limited adoption of rigorous practices.

**Position:**

Yes

**Position In Title:**

Yes

**Related Work:**

3

**Strengths And Weaknesses:**

Strengths：
High significance and strong alignment with community pain points: Issues such as benchmark contamination, irreproducible results, and unstable evaluations can systematically distort conclusions about model capabilities. Framing rigor, reliability, and reproducibility as preconditions for trustworthy benchmark claims is a defensible and discussion-worthy position.
	Actionable, process-oriented perspective: Organizing quality practices around the benchmark lifecycle—design / construction / evaluation / analysis / release—is more executable than scattered recommendations, and the tiered prioritization (high/medium/low priority) has practical engineering value for real-world adoption.

Weaknesses：
Potential non-compliance with the Position track “Alternative Views” requirement: The opposing viewpoints are presented in a fragmented manner, rely on limited supporting evidence, and the response lacks a concrete compromise plan or implementable policy, creating a compliance risk.
	Insufficient validation of the central motivation (broken causal chain): The paper convincingly shows that quality gaps are prevalent, but provides limited direct evidence that fixing these gaps would materially change benchmark conclusions, rankings, or downstream research directions; thus the strongest causal claims remain largely inferential.
	Inconsistency in core statistics (QA checks): The reported proportions regarding whether benchmarks underwent quality assurance checks appear inconsistent across the paper and are difficult to reconcile without clarifying definitions or denominators. In a paper advocating rigor, this undermines confidence in the broader set of empirical claims.

**Support:**

2

---

> ### Author Rebuttal · Authors · 2026-03-31
>
> ## Q1. Inconsistent statistics for QA check:
>
> Sorry for the typo. The text:
>
> > only 31% went through a QA check
>
> should be corrected to:
>
> > only **48.1%** went through a QA check
>
> ## Q2. QA check criteria
> Below shows the criteria for the annotation for QA check.
>
> - **Criteria for Checking whether Data quality assurance present**
>     > **Criteria**: We labeled a benchmark as "considers data quality assurance" only if we identified a dedicated section, subsection, or explicit sentences describing how data quality was ensured. In the absence of such textual evidence, we labeled the benchmark as not considering data quality assurance.
>
>     Annotators look for explicit discussion of how data quality is ensured. Typical evidence includes sections or sentences tagged as **"Data Curation / Cleaning / Preprocessing", "Quality Control / Quality Assurance"**, etc., or detailed descriptions of construction and filtering procedures.
>
> - **Criteria for Type of data quality assurance: Manual check**:
>
>     > **Criteria**: We mark a benchmark as using manual checks if it explicitly mentions human involvement in inspecting or validating data/labels, with **keywords** such as:
>
>     - "manual", "human", "inspection", "verification", "review"
>     - "spot check", "random sampling", "audt", "sanity check"
>     - "hand‑curated", "human‑in‑the‑loop", "expert"
>     - "label correction", "error analysis", "case‑by‑case"
>
> - **Criteria for checking whether Type of data quality assurance: Code execution check**
>     > **Criteria**: We mark a benchmark as using code execution checks if it describes validating items by actually running code or tests, with **keywords** such as:
>     - "execute/execution", "run", "compile", "build"
>     - "unit tests", "test suite"
>     - "pass/fail"
>
> - **Criteria for checking Type of data quality assurance: LLM‑as‑judge**:
>
>     > **Criteria**: We mark a benchmark as using LLM‑as‑judge if it delegates evaluation or filtering to a model as a judge, with phrases like:
>
>     - "LLM‑as‑a‑judge", "LLM judge"
>     - "model judge"
>     - "[model name] as judge"
>
> Q3. Representative works
>
> The selection criteria for the focused case study is explained in Section 3. Step 3: “Focused Case Study. After obtaining an overall impression of existing benchmarks, we selected 30 (= 5 * 6) representative benchmarks from top-5 tasks, with top-5 highly-cited benchmarks plus the latest 1 benchmark (Appendix E).” The cut-off date of these statistics is 31/8/2024, as noted in Appendix E.
>
> Revision:
>
> To ensure the timeliness, we updated the focused case study by setting the cut-off date as 31/12/2025, replacing 4 benchmarks, adding 5 benchmarks,  detailed as follows:
>
> - Code Generation: Add CIFE [1] (replacing VerilogEval)
> - Defect Detection: Add Pushkar et al. [2] (replacing VulDetectBench)
> - Program Repair: Add BackPortBench [3] (replacing DebugBench)
> - Code Summarization: Add ContextCRBench [4] (replacing  Long Code Arena)
> - Code Reasoning: LiveCodeBench [5], CRUXEval [6], xCodeEval [7], CodeQA [8], RepoQA [9]. Latest  one as of 31/12/2025: RE2-Bench [10]
>
> Key statistics after update:
>
> - Appendix C.2: 80% of benchmarks explicitly specify the capabilities (e.g., intention understanding, problem solving, testing, and debugging capabilities)to be evaluated, while 20% do not.
> - Appendix C.2: 76.7% representative benchmarks have clearly specified application scenarios (e.g., programming assistant), while the rest do not.
> - Appendix C.4: 76.7% representative benchmarks do not validate whether the prompts they used are well-designed
> - Appendix C.5: 63.3% benchmarks have detailed explanations and analyses of their evaluation results, while still 36.7% have not.
>
> ## Q4. Criteria for data contamination check
>
> See the answer to Q1 for https://openreview.net/forum?id=xKjGSyryju&noteId=c5Ct4HHCBW (Reviewer qktb, the last reviewer).

---

> > ### Author Rebuttal · Reviewer_cFaa · 2026-04-04
> >
> > Thank you for your response. I have no further questions. I will consider adjusting my score in the subsequent stage.

---

### Official Review · Reviewer_4TJY · 2026-03-12

**Significance:** 4
**Argument Clarity:** 4
**Rating:** 5
**Confidence:** 4

**Questions:**

Questions and minor suggestions:
1. What is snowballing?
2. Clarify what top-5 tasks means (clarified in Appendix, needs to be clear in the main body)
3. Broken formatting line 308 (9.9= 48 / 485). Also that is only 10% of benchmarks, what about the rest?
4. When logs are not available, is this due to licensing issues, practicality of anonymizing and publishing logs, general inability to release them, or other issues? Was there any effort to contact authors of the 30 representative papers?
5. Some of these categories are obvious/highly subjective. What prevents a designer from simply self-judging that they have satisfied categories if there are no clear guidelines (specially under the first section of categories)
6. How common are the out of capability cases? If a benchmark has 1K datapoints and 1 is poorly formed, then it is not a big concern
7. A discussion would be useful on the effectiveness of these criteria depending on whether the benchmark is built for training vs evals

**Alternative Views Section:**

Yes

**Compliance With Llm Reviewing Policy A Conservative:**

Affirmed.

**Discussion Potential:**

4

**Final Justification:**

I appreciate the authors addressing my concerns and the weaknesses I mentioned. Given their responses, I feel my current score is justified, even considering the other reviews. I therefore maintain my score.

**Paper Summary:**

The authors of this paper present the position that code benchmarks should prioritize rigor, reliability, and reproducibility. They start by exploring issues in existing code benchmarks, surveying more than 550 benchmark papers, as well as other evaluation papers. They demonstrate that this is an important problem that has just recently started receiving some awareness, and develop a 55-category checklist to evaluate the quality of each benchmark. Finally they conduct a user study to evaluate the criteria they developed and obtain feedback.

**Position:**

Yes

**Position In Title:**

Yes

**Related Work:**

4

**Strengths And Weaknesses:**

Strengths:

The paper is well motivated and presented, basing their position on concrete, well-established evidence. This is also a real problem in a very actively researched space, with implications in the real world as more code starts getting generated via LLMs. I am impressed by the scale of the survey (550+ benchmarks), as well as the thought put into developing the criteria. I also appreciate the substantive analysis of existing benchmarks and trends displayed over time.

Weaknesses:
I only see two significant issues in this paper:
1. The alternative view section does not really look at alternative views: it mainly describes an additional dimension that needs to be considered. No one outright said rigor of code eval benchmarks is bad, and the exploration into the views provided by the user feedback is not thorough. I would appreciate more discussion here.
2. The paper does not discuss the tradeoff between reproducibility and the need to keep benchmarks hidden to prevent data leakage issues. Some benchmark papers achieve this by encrypting the benchmark itself when it is publicly released. I also wanted to see discussions of hidden datasets for leaderboard evals and the tradeoff between trusting their evals and risking future LLM versions training on them.
3. There is no discussion of how practical it is to achieve these criteria and the associated costs, neither is there discussion on methods one can use to improve the benchmarks based on the criteria.

Some other minor issues:
1. The authors mention that they interviewed people to develop the criteria, but do not expand on the interview process, details about the interviewees, or the feedback obtained from them.
2. Awareness of code quality has only grown in recent years as per the authors' survey, but they do not show how much the quality has improved just in that timeline; the current analysis shows issues in code quality in benchmarks from 2015--2025.
3. A dimension that is not considered is the size of these benchmarks: some may be a few hundred samples, while others may have thousands
4. Re. awareness, it is important to split the analysis between people who have created benchmarks and those who haven't; I would be significantly more concerned if the 15% of people who were unaware of data denoising were people who have _successfully_ published a well cited benchmark
5. How are priorities established? I am surprised that ensuring code compiles and runs is optional, especially when a large number of code gen benchmarks evaluate code by running tests on it

**Support:**

3

---

> ### Author Rebuttal · Authors · 2026-03-31
>
> ## Q1. What is snowballing?
>
> In the context of a literature review, snowballing is a search strategy where one uses already‑identified relevant papers to iteratively discover additional relevant work, analogous to a snowball growing in size as it rolls.
>
> In our study, we used both:
>
> - **Backward snowballing** (reference snowballing):
> Starting from a seed paper, we examined its reference list and selected additional benchmark papers that met our inclusion criteria.
>
> - **Forward snowballing** (citation snowballing):
> Starting from a seed paper, we used Google Scholar to find newer papers that cite this seed paper, and again selected relevant benchmarks.
>
> In our implementation, Google Scholar was the main platform used for forward snowballing. We will add this explanation to Section 3, Step 2 (around line 194) to make the procedure clearer.
>
> ## Q2. Clarify what “top‑5 tasks” means
>
> “Top‑5 tasks” (line 206) refers to the five most common coding task types among the benchmarks we surveyed, as shown in Figure 9.
>
> We will clarify this at first mention:
> > (line 206) “... from top‑5 most frequent coding task types (see Figure 9) and ...”
>
> ## Q3. Broken formatting at line 308 and coverage of the rest
>
> Thank you for catching the formatting issue. The text:
> > “(9.9= 48 / 485)”
>
> should be corrected to:
>
> > “(9.9% = 48 / 485)”.
>
> We also agree that the current phrasing under‑reports what happens in the rest of the distribution.
>
> The complete statistics are:
>
> - 9.9% of benchmarks (48/485) were evaluated on exactly 3 models.
> - This is followed by 8.9% (43/485) evaluated on 6 models, and 7.8% (38/485) evaluated on 7 models.
> - 63.7% of benchmarks were evaluated on 1–10 models.
> - On the other hand, 5.4% (26/485) were evaluated on 30 or more models.
>
>
> ## Q4. Log availability
>
> Regarding author contact, we did not contact the authors of 30 representative benchmarks to request logs or other information, because our analysis is based on what is publicly and readily available (papers, repositories, and documented artifacts).
>
> We agree that the reviewer has mentioned reasonable and practical reasons for this issue, and it is worthy mentioning in the paper. We will clarify in the final version that:
>
> > Our conclusions about log availability are based on publicly accessible materials,
> Logs are often missing for a combination of policy, privacy, and practicality reasons.
>
> ## Q5. Subjectivity of the category
>
> We appreciate the concern that some categories may appear obvious or subjective.
>
> In practice, such situations are indeed possible and, to some extent, unavoidable: problems in benchmark design are often latent and only become apparent ex post, for example when (i) almost all models systematically fail on certain items, or (ii) performance on a benchmark is clearly inconsistent with performance in realistic deployment settings.
>
> We share this concern and will explicitly discuss it at the end of Section 4.1. In particular, **we will caution practitioners that our checklist should not be read as “blindly trust whatever the benchmark designer claims,”** but rather as a set of aspects that users should actively scrutinize, including the quality and credibility of the evidence provided by the benchmark designer for each claimed category.
>
>
> ## Q6. How common are out‑of‑capability / poorly formed cases?
>
> It is correct that an isolated problematic item in a very large benchmark (e.g., 1 poorly formed item out of 1K) is usually not a major concern.
>
> Our goal was not to overstate this as a pervasive or dominant issue.
>
> Based on our close examination of the 30 representative benchmarks:
>
> - **Out‑of‑capability / poorly formed cases are not extremely frequent**: they do occur, but not at a rate that would by itself invalidate most benchmarks.
> - When they do occur, they tend to be clustered in particular sub‑areas (e.g., tasks that rely on very specific domain knowledge or ambiguous instructions) rather than uniformly spread across all items.
>
> In the paper, we highlighted such cases mainly as an illustrative example of a failure mode that benchmark designers should watch for, not as evidence that existing benchmarks are overwhelmingly dominated by such issues.
>
> In the revision, we will:
>
> - Explicitly state that these cases are relatively rare in our 30 focused benchmarks and are presented as instructive examples rather than primary failure modes.
> - Clarify that the impact depends on both frequency and placement (e.g., whether problematic items disproportionately affect particular sub‑scores or heavily used leaderboards).
>
>
> ## Q7. Effectiveness of criteria for training vs evaluation benchmarks
>
> This is an excellent point. The importance and interpretation of our criteria indeed differ depending on whether a benchmark is primarily used for evaluation or is repurposed as training/val datasets.
>
> The ratio of benchmarks built for training/val is 23.3%.
>
> We will add a dedicated discussion section to make this distinction explicit.

---

> > ### Author Rebuttal · Reviewer_4TJY · 2026-04-02
> >
> > Can you please address my concerns under the weaknesses section? Please address all the three major weaknesses and how you plan to address it here, or whether addressing it needs a major revision. I am seeing similar concerns across other reviews, and might be forced to reduce my score if not addressed.
> >
> > Also address the issues from the minor weaknesses, at the very least points 4 and 5.

---

### Official Review · Reviewer_U3hL · 2026-03-13

**Significance:** 2
**Argument Clarity:** 3
**Rating:** 4
**Confidence:** 2

**Questions:**

1. Many statistics rely on manually identifying whether benchmarks follow certain practices. More details are needed on how these attributes were annotated, whether multiple annotators were involved, and whether any inter-annotator agreement was measured.

2. In Section 4.2, the paper discusses the importance of code coverage and test coverage for benchmark test cases. However, it is unclear how these coverage metrics are defined and measured in the context of this study. Is the coverage computed with respect to the reference implementation, generated code, or another form of ground-truth program? Are  "code coverage" and "test coverage" the same concept or have distinct connotations? Additionally, how did the authors determine whether a benchmark "considers coverage" when analyzing the 572 benchmarks?

3. The paper proposes a guideline, but its practical effectiveness is not fully demonstrated. Have the authors applied the checklist to construct or revise a benchmark and compared the results before and after applying the guideline?

**Alternative Views Section:**

Yes

**Compliance With Llm Reviewing Policy A Conservative:**

Affirmed.

**Discussion Potential:**

2

**Final Justification:**

The rebuttal has addressed my concerns and I have changed my score to borderline accept.

**Paper Summary:**

This paper investigates the methodological quality of code-related benchmarks used to evaluate LLMs. The authors observe that although awareness of benchmark quality has increased in recent years, it remains unclear whether this awareness has translated into better benchmark construction practices. To study this issue, the paper conducts a large-scale survey of 572 code benchmarks, systematically examining their design, construction, evaluation, analysis, and release processes.

Based on this analysis, the authors identify widespread methodological shortcomings in existing benchmarks. To address these issues, the paper proposes HOW2BENCH, a guideline consisting of 55 checklist items that cover the full lifecycle of benchmark development, aiming to improve rigor in benchmark construction, reliability in evaluation, and reproducibility in benchmark release. The study combines statistical analyses of the surveyed benchmarks, case studies of representative benchmarks, and a human study to support these findings and highlight gaps in current benchmark practices.

**Position:**

Yes

**Position In Title:**

Yes

**Related Work:**

3

**Strengths And Weaknesses:**

Strengths:

+ Important and timely problem. The paper studies the methodological quality of LLM code benchmarks, which is highly relevant as such benchmarks are widely used to evaluate model capabilities and influence research conclusions.

+ Large-scale and systematic empirical analysis. The study analyzes 572 benchmarks, providing a broad and data-driven view of current benchmark construction and evaluation practices.

+ Practical contribution to the community. The proposed HOW2BENCH checklist offers concrete guidelines that can help researchers improve rigor, reliability, and reproducibility when building future benchmarks.

Weaknesses:

- The HOW2BENCH checklist lacks empirical validation. Although the checklist is comprehensive, the paper does not clearly demonstrate, through controlled experiments or concrete case studies, how applying the checklist would significantly improve benchmark quality or evaluation reliability in practice.

- Some analyses remain descriptive rather than causal. Many findings are based on statistical observations, but the paper provides limited deeper analysis of how these issues concretely affect evaluation outcomes or model ranking.

**Support:**

2

---

> ### Author Rebuttal · Authors · 2026-03-31
>
> ## Q1
> See the answer to Q1 for https://openreview.net/forum?id=xKjGSyryju&noteId=c5Ct4HHCBW (Reviewer qktb, the last reviewer) where:
>
> - Annotation details
> - Inter-annotator scores
> - Criteria for the annotation
>
> are explained.
>
> ## Q2. Definition and measurement of coverage; “code coverage” vs “test coverage”
>
> Thank you for pointing out this ambiguity. Sorry for the confusion caused, and we will revise the text for clarity.
>
> **Terminology correction**
>
> In the current draft, the term "test coverage" is misused and **should be replaced by "code coverage" throughout**. Specifically:
>
> 1. We will replace the six occurrences of “test coverage” with “code coverage” in the paper.
>
> 2. In our paper, we do not define a separate metric called "test coverage"; all coverage‑related discussion refers to how much of the reference implementation’s source code is executed by the provided test suite.
>
> **What code coverage refers to in our context**
>
> In our study, when we talk about *coverage*, we mean:
>
> > **Code coverage**: The proportion of the lines/branches/statements in the reference implementation that are executed at least once when running the official tests provided by the benchmark.
>
> Concretely:
>
> 1. The ground truth program is the reference implementation shipped (or implied) by the benchmark (i.e., the canonical solution used to generate expected outputs or to validate correctness).
> 2. Coverage is not computed with respect to model‑generated code; rather, we reason about coverage hypothetically with respect to the reference implementation.
> 3. The core point in Section 4.2 is conceptual: if the reference tests do not sufficiently exercise different branches or corner cases in the reference implementation, then it becomes easier for incorrect or partially correct generated code to pass these tests, leading to inflated or unreliable evaluation scores.
>
> Where we provide concrete examples in the paper, “code coverage” can be interpreted as line or branch coverage over the reference implementation; we will clarify this explicitly in the final version.
>
>
> **How we determine whether a benchmark “considers coverage”**
>
> When analyzing the 572 benchmarks, we did not re‑implement or run coverage tools on all benchmarks (which would be infeasible at this scale). Instead, we examined whether each benchmark:
>
> - Explicitly defines or measures code coverage, OR
> - Explicitly discusses coverage‑related aspects of its test suite (e.g., coverage of corner cases, branch/statement coverage, test adequacy metrics), OR
> - Uses terms that clearly indicate an intentional design to exercise a broad range of program behaviors.
>
> Concretely, annotators searched for **textual evidence** such as:
>
> - *Direct mentions*: “code coverage”, “branch coverage”, “line coverage”, “statement coverage”, “coverage metric”, “coverage‑guided”
> - *Indirect but related phrasing*: “test adequacy”, “test strength”, “test suite quality”, “ensure that all branches are tested”, “cover edge/corner cases” when explicitly tied to systematic coverage rather than anecdotal examples.
>
> A benchmark is labeled as considering coverage only if:
>
> - Such terms appear in the context of test design or evaluation, and
> - The text indicates that the authors intentionally use coverage (or coverage‑like criteria) to design, evaluate, or justify their test suite.
>
> We will clarify this process in the final version in Appendix.
>
> ## Q3. Practical effectiveness of the checklist / guideline
>
> We agree that demonstrating the practical effectiveness of the checklist with a before‑and‑after study would further strengthen the paper.
>
> In the current version, we have not constructed a new benchmark from scratch using the checklist. We will clarify this in the text to better set expectations for the contribution and scope, especially given the “position paper” track.
>
> **Planned follow‑up and partial evidence**
>
> We agree with the reviewer that a stronger empirical of revising existing benchmarks to address specific weaknesses (e.g., adding contamination checks, improving coverage, adding deduplication or manual checks) and re‑evaluating model performance before and after revision would be highly valuable.
>
> Due to time and space constraints, it's hard to provide experimental result during rebuttal. If permitted in the camera‑ready, we will add such experiment to show how applying parts of the checklist can change the apparent difficulty and reliability of a benchmark.

---

> > ### Author Rebuttal · Reviewer_U3hL · 2026-04-03
> >
> > I appreciate the authors' efforts in their response. The response fully addressed my concerns, and I am happy to raise my score to borderline accept.

---

### Decision · Program_Chairs · 2026-04-30

**Decision:**

Accept (regular)

**Comment:**

The authors show a critical gap and advocate a clear position. Overall, the referees are also positive about the paper. However, the checklist proposed remains to be validated. Also, the alternative views section needs to be worked on massively.